# Persistent activity in human parietal cortex mediates perceptual choice repetition bias

**Anne E. Urai** [ID][1,2,3] [✉] **& Tobias H. Donner** [ID][1,2,4] [✉]

Humans and other animals tend to repeat or alternate their previous choices, even when judging sensory stimuli presented in a random sequence. It is unclear if and how sensory, associative, and motor cortical circuits produce these idiosyncratic behavioral biases. Here, we combined behavioral modeling of a visual perceptual decision with magnetoencephalographic (MEG) analyses of neural dynamics, across multiple regions of the human cerebral cortex. We identified distinct history-dependent neural signals in motor and posterior parietal cortex. Gamma-band activity in parietal cortex tracked previous choices in a sustained fashion, and biased evidence accumulation toward choice repetition; sustained beta-band activity in motor cortex inversely reflected the previous motor action, and biased the accumulation starting point toward alternation. The parietal, not motor, signal mediated the impact of previous on current choice and reflected individual differences in choice repetition. In sum, parietal cortical signals seem to play a key role in shaping choice sequences.

The tendency to systematically repeat or alternate choices is ubiquitous in decision-making under uncertainty. Repetition effects in choice sequences have been called decision inertia[1] or perseveration bias[2]. The terms intertrial dependence[3], sequential choice bias[4] and choice history bias[5–8] more generally refer to both repetition and alternation tendencies in choice sequences. Such choice history biases are prevalent even when observers judge weak sensory stimuli presented in a random sequence[9–11], and they generalize from humans[1,3,6,12] to monkeys[13–15] and rodents[8,16–22]. Choice history biases can be adapted to the correlation structure of stimulus sequences[5,6,8], and they depend on the previous decision's confidence[4,6,12,22].

Choice history biases seem to be shaped by sensory and motor, but in particular by central processing stages. Sensory stimuli[6,8,20,23] and motor responses[24] both tend to repel subsequent choices. Yet, reporting a choice with the same or different motor act seems to have little effect on choice history biases[1,6,25,26]. Behavioral analyses suggest that the biases are dominated by observers' previous perceptual interpretation of the sensory input, rather than the sensory input or motor act per se[1,6,23,25].

Neural signatures of previous choices have been identified in primate visual cortex[14,27], although their causal role has been disputed[28]. The state of human motor cortex reflects previous motor acts[24,29,30] and, in certain task protocols, predicts an individual's tendency towards choice alternation[24]. In animals, associative areas of posterior parietal cortex[18–20,31] and prefrontal cortex[15,32] carry history information that seems to play a causal role in choice history biases. Critically, however, previous studies of the neural bases of choice history bias have commonly focused on a single brain region. Thus, it has remained unclear how neural history signals in different (sensory, associative, or motor) cortical areas conspire to shape behavioral choice sequences.

Here we disentangled neural choice history signals across the visuo-motor cortical pathway, and linked them to distinct dynamic computations underlying the formation of a visual decision. We analyzed MEG and behavioral data from 60 observers who discriminated small changes in visual motion strength. We delineated multiple neural signals reflecting choice and action history, expressed in different cortical areas and frequency bands. Notably, choice history biases in

[1]Section Computational Cognitive Neuroscience, Department of Neurophysiology and Pathophysiology, University Medical Center Hamburg-Eppendorf, Hamburg, Germany. [2]Department of Psychology, University of Amsterdam, Amsterdam, The Netherlands. [3]Cognitive Psychology Unit, Institute of Psychology, Leiden University, Leiden, The Netherlands. [4]Bernstein Center for Computational Neuroscience, Berlin, Germany. [✉]e-mail: a.e.urai@fsw.leidenuniv.nl; t.donner@uke.de

the face of random stimulus sequences are highly idiosyncratic, leading some observers to systematically alternate, and others to systematically repeat their choices[7,12]. Thus, we also asked whether any of the so-identified neural history signals mirrored these idiosyncratic behavioral patterns. We found that gamma-band activity in higher-tier areas of posterior parietal cortex mediated idiosyncratic biases towards choice repetition, but not alternation.

## Results

Participants ($n = 60$) performed a two-interval motion coherence discrimination task (Fig. 1a). Throughout each trial, observers viewed dynamic random dot patterns of varying motion coherence. In two successive intervals, called "reference" and "test" stimulus (onset of each stimulus cued by an auditory beep), some dots moved in one of the four diagonal directions (fixed per observer). The reference coherence was 70% in all trials. The test coherence differed (toward stronger or weaker) from 70% by a small amount that yielded a threshold accuracy (about 70% correct) for each individual (see "Methods"). Observers judged whether the coherence of the test was stronger or weaker than that of the reference. Auditory feedback was presented after a variable delay.

As observed previously[7,12], choice behavior in this task revealed stable, idiosyncratic choice history biases (Fig. 1b–d). By design, stimulus categories (test stronger vs. weaker than reference) were largely uncorrelated across trials (Fig. 1b, left). The autocorrelation in choice sequences was considerably larger, with observers ranging from alternating to repeating their past choices (Fig. 1b, right). In line with previous reports[12,33], these individual differences were relatively stable across two experimental sessions 13–30 days apart (Fig. 1c). For further analyses, we divided participants into two sub-groups based on their choice repetition probability collapsed across both sessions (Fig. 1d). One observer had a repetition probability of exactly 0.5, and was

excluded from subgroup analyses. In what follows, we label these two sub-groups as "repeaters" ($N = 34$) and "alternators" ($N = 25$), without implying a statistically significant deviation from 0.5 within each individual of these groups. Yet, repetition probabilities differed from 0.5 so consistently across blocks that they were statistically significant ($p < 0.05$, $t$ test) in a substantial number ($N = 12$ each) of individuals from each group (large symbols in Fig. 1d). Detecting more subtle individual deviations from 0.5 may require more within-subject data.

Choices from multiple past trials contributed to the history biases (Fig. 1e, f). Regression modeling uncovered an effect of choices made more than one trial back, specifically for repeaters (Fig. 1e). Notably, repeaters and alternators were solely differentiable based on the kernels quantifying the impact of previous choices (Fig. 1e, compare orange and purple), rather than of previous stimuli or of combinations of both (e.g., win-stay/lose-switch strategy; Supplementary Fig. 1a, b). In line with previous reports[1,8], we also observed a build-up of the history bias across multiple trials, forming a streak of the same choice (Fig. 1f). This was evident in an interaction between sequence length and final choice: when streaks ended in a single alternation, repetition biases disappeared or even reversed (Fig. 1f). This interaction was specifically expressed in repeaters, but not in alternators (Fig. 1f, with a significant group difference: mixed model, interaction *sequence length × sequence end × subgroup* F(4) = 5.664, $p < 0.001$; "Methods"). These streak effects were independent of trial outcomes (Supplementary Fig. 1c).

### Stimulus- and action-related dynamics across the visuo-motor pathway

To pinpoint neural signatures of choice history bias in our task, we focused on established MEG signatures of visual motion processing and action planning, within well-defined cortical regions and frequency bands[30,34–38]. We first replicated these signatures in our current

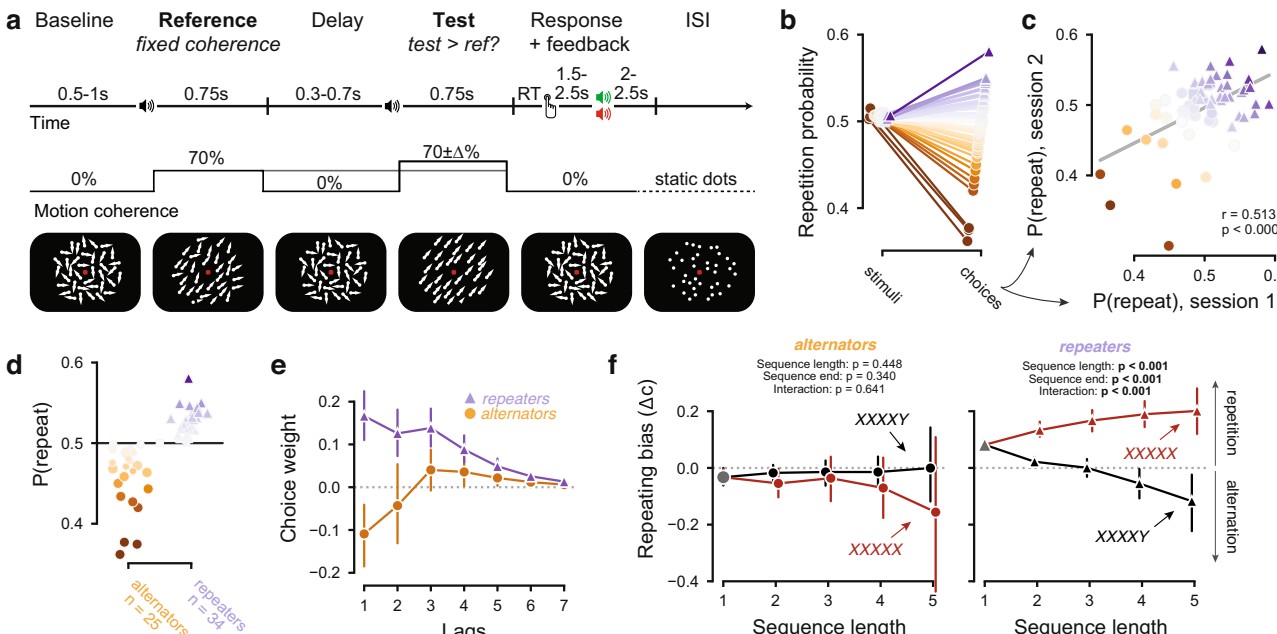

**Fig. 1 | Behavioral task and choice history biases. a** Behavioral task. **b** Probability of repeating the previous stimulus category (left) or choice (right). **c** Correlation between individual P(repeat) in the first and second MEG sessions (Pearson's $r = 0.5134$, $p = 0.00003$). **d** Choice repetition separately for alternators ($N = 25$, orange circles) and repeaters ($N = 34$, purple triangles). One observer had a repetition probability of exactly 0.5, and was excluded from subgroup analyses. Large markers indicate those individuals whose block-wise repetition probability was significantly different from 0.5, as determined by a $t$ test. **e** Impact (regression weights) of several previous choices on current bias. $T$ test against 0 within each

subgroup and lag: significant for repeaters on lags 1–7; significant for alternators on lag 1. **f** Build-up of history bias across multi-trial choice streaks (i.e., successive repetitions of same choice), for both groups. Sequences were separated by whether they end in a repeating (red purple) or an alternating (black) choice. "Repeating bias" was quantified as shift in decision criterion into the direction of final choice (i.e., "X" for red sequences, "Y" for black sequences). Main effects (Sequence length: $p = 4.14e-8$, sequence end: $p = 1.71e-7$) and interaction ($p = 1.14e-25$) from a repeated-measures ANOVA (see "Methods"). Data are shown as mean +/− 95% bootstrapped confidence intervals ($n = 60$).

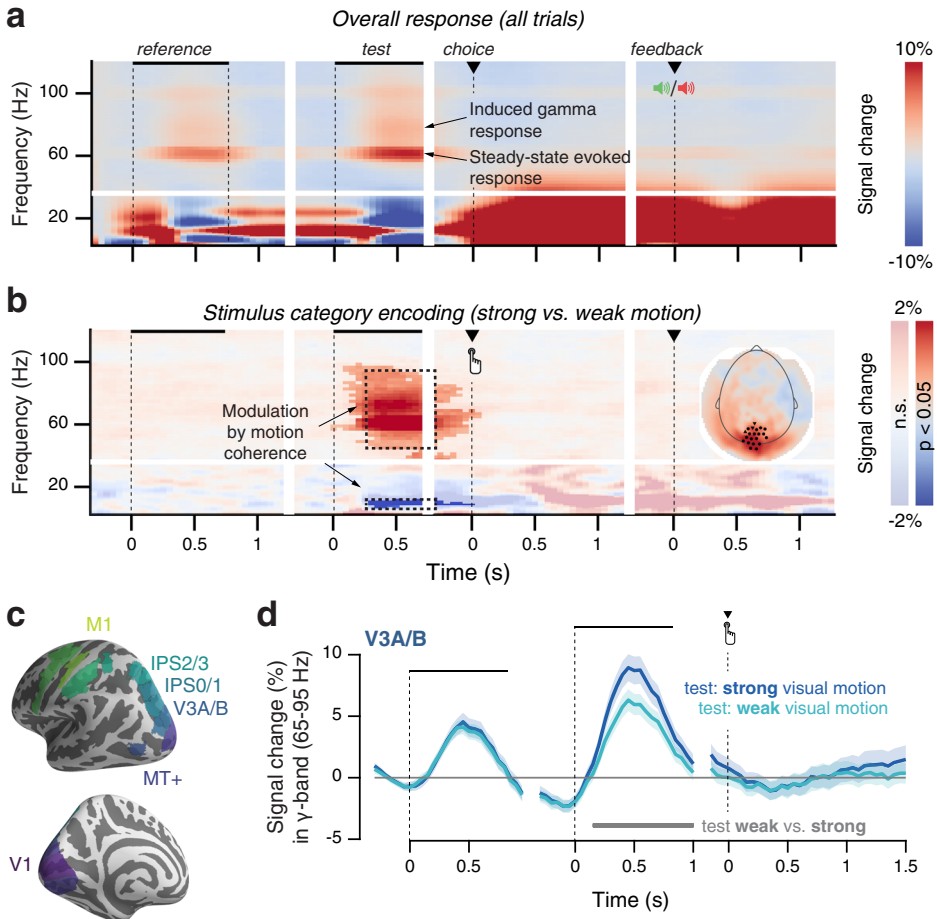

**Fig. 2 | Task-related MEG dynamics in visual cortex.** Time–frequency representation of MEG power modulations relative to pre-trial baseline **a** across trials, or **b** contrasting task-relevant stimulus categories (strong vs weak motion). Inset: occipital sensors, selected based on the visual motion-dependent gamma-band response. Shading indicates significant clusters in time-frequency space, as determined by a cluster-based permutation test on preselected sensors (see "Methods"). **c** Regions of interest for source reconstruction, displayed on the inflated cortical surface. **d** Gamma-band activity in motion-selective visual areas, such as V3A/B, scales with the strength of coherent motion presented on the screen. Data are shown as mean +/− 95% confidence intervals ($n = 60$).

measurements (Figs. 2 and 3). This laid the ground for delineating, and functionally characterizing, choice history signals within the same areas and frequency bands.

During both intervals containing coherent motion (i.e., reference and test), we observed occipital enhancement of high-frequency power (from 30–100 Hz, including a steady-state response at 60 Hz, Supplementary Fig. 2), which was accompanied by a suppression of low-frequency power (<30 Hz) (Fig. 2a). Our subsequent analyses focused on the modulations of high gamma band power (65–95 Hz), which may reflect broadband population spiking[36,39,40]. As in previous work[34], we found enhanced visual gamma-power responses and alpha-power suppression to stronger versus weaker motion coherence (Fig. 2b), in visual cortical areas known to be involved in visual motion processing, such as area V3A/B (Fig. 2d, "Methods"). Thus, both signals tracked the subtle sensory signal relevant for the near-threshold discrimination task.

Also in line with previous work[24,35,38], in motor cortex we observed gradual build-up of the lateralization of alpha- and beta-band power suppression contralateral vs. ipsilateral to the upcoming response. This signal ramped up during decision formation, from the test interval up until the execution of the button press (Fig. 3a). Then, the signal flipped from contralateral suppression to contralateral enhancement ("beta rebound", ref. 42), an effect that carried over to the next trial (Fig. 3b, c) and was most prominent in the hand area of primary motor cortex (M1, Fig. 3c; cf. ref. 24).

## Choice history signals in parietal cortex
Having replicated these established spectral signatures of sensation and action at sensory and motor processing stages of our task, we next sought to identify history-dependent neural signatures across a number of precisely delineated cortical areas covering the visuo-motor pathway[37,38]: a hierarchically ordered set of dorsal visual field maps (from V1 into intraparietal cortical area IPS2/3), plus parietal and frontal regions carrying action-selective activity (Table 1; Fig. 2c).

The stimulus category (stronger vs. weaker motion) modulated high gamma-band responses in all visual field maps up to IPS0/1 (Fig. 4a, top). Critically, in IPS2/3, gamma-band activity tracked the previous trial's choice, being enhanced after a "stronger" choice, both in the reference (Fig. 4b) and test intervals (Fig. 4c). This effect was only present in repeaters but not for alternators, and differed significantly between groups of subjects (Fig. 4d, top). The effect in repeaters was present even when randomly subsampling 25 observers (the same number as in the group of alternators: effect of previous choice = 0.919, CI [0.363, 1.475], $p = 0.001$), and when tested only in the 12 repeaters whose repetition probabilities significantly differed from 0.5 (effect of previous choice = 1.0686, CI [0.25196, 1.8853], $p = 0.0103$).

Likewise, choice history affected alpha-band power in the same direction in neighboring area IPS0/1 (the first in the visual hierarchy where the stimulus category did not modulate alpha-band power; Fig. 4a, bottom), but only during test (Fig. 4c, bottom) and not during

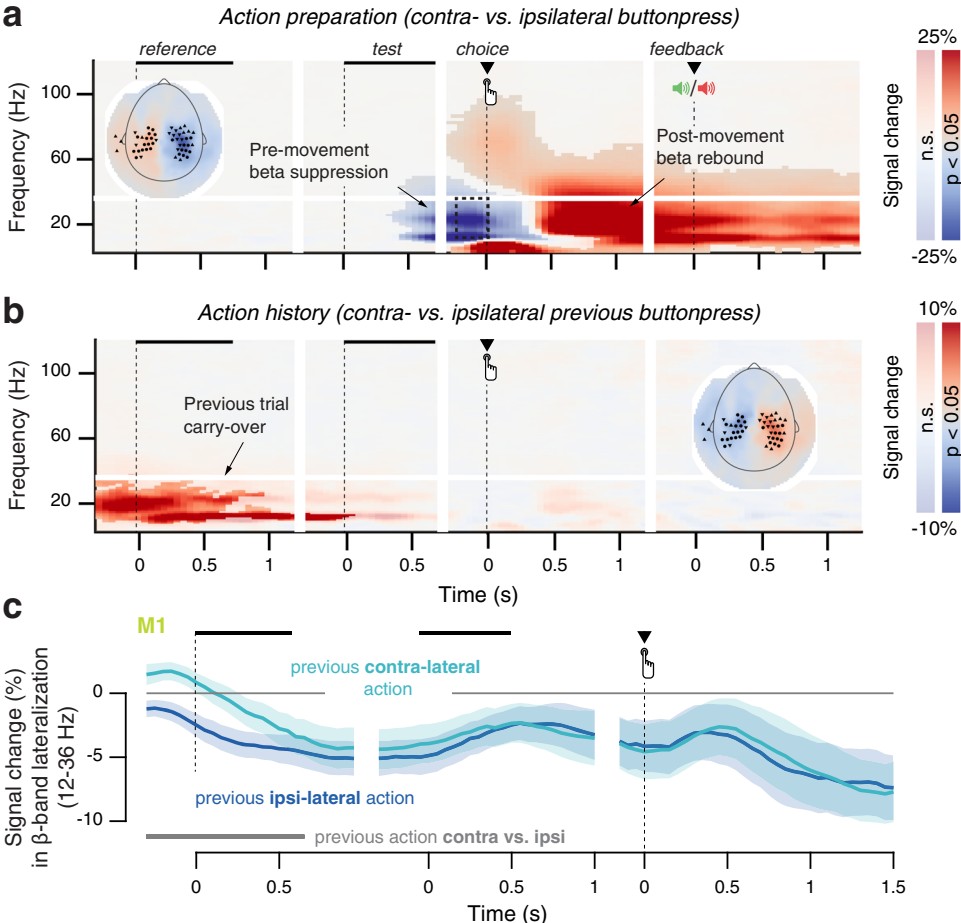

**Fig. 3 | Action-related MEG dynamics in motor cortex. a** Time–frequency representation of action-related MEG power lateralization (contralateral vs. ipsilateral to button press) over motor cortex. Inset: motor sensors, selected based on pre-movement beta-band lateralization. Shading indicates significant clusters in time-frequency space, as determined by a cluster-based permutation test on preselected sensors (see "Methods"). **b** As **a** but now lateralization contralateral vs. ipsilateral to the previous motor action. **c** Time course of beta-band (12–36 Hz) lateralization in the hand area of M1, separately for previous contra- and ipsilateral motor actions. Data are shown as mean +/− 95% confidence intervals (n = 60).

reference (Fig. 4b, bottom; only effect during this interval in V2–4). Differently from the IPS2/3 gamma signal, the IPS0/1 alpha signal did not distinguish between alternators and repeaters (Fig. 4d, bottom). The alpha signal exhibited a superposition of two signals: a specific enhancement of alpha-power in IPS0/1 after incorrect "stronger" choices (Supplementary Fig. 3b, bottom) and a global, error-related alpha-suppression unrelated to the previous choice (Supplementary Fig. 3c, bottom). For all subsequent analyses, we used an isolated measure of the choice history-specific component (Methods).

### Table 1 | Region of interest definition

| Cluster | Functional areas | Source |
|---|---|---|
| V1 | Dorsal and ventral parts of V1 | ref. 54 |
| V2–V4 | Dorsal and ventral parts of V2, V3, V4 | |
| V3A/B | V3A, V3B | |
| MT+ | MT, MST | |
| IPS0/1 | IPS0, IPS1 | |
| IPS2/3 | IPS2, IPS3 | |
| aIPS | aIPS1 | ref. 55 |
| IPS/PostCeS | IPS/post-central sulcus | |
| M1 | M1 (hand area) | |
| PMd/V | 55b, 6d, 6a, FEF, 6v, 6r, PEF | ref. 78 |

Several patterns in the two parietal history effects, expressed in IPS2/3 gamma and IPS0/1 alpha, suggests distinct underlying processes. First, the effect of previous choices on IPS2/3 gamma-band power in repeaters was sustained throughout the trial (Fig. 5a). By contrast, the IPS0/1 alpha history signal (Fig. 5b) first emerged transiently during the processing of that decision (i.e., test stimulus interval; Figs. 5 and S1, left) and then re-emerged during the inter-trial interval (Supplementary Fig. 5b). Second, both parietal history signals showed an opposite dependency on the previous outcome, with the IPS2/3 gamma effect only present after correct trials, and the IPS0/1 alpha effect only after error trials (compare Supplementary Figs. 3a and S3b). Third, and most importantly, only the IPS2/3 gamma-band effect differentiated between subgroups of repeaters and alternators.

There was a tendency for the IPS2/3 gamma-band effect to build up over streaks of successive repetitive choices (Supplementary Fig. 4b), similar as observed for the behavior (Fig. 1f). Specifically, there was a significant interaction between sequence length and sequence end (repetition vs. alternation) on build-up of IPS2/3 gamma-band, for repeaters but not alternators (Supplementary Fig. 4b). In other words, also the across-trial behavior of the history signal in IPS2/3 gamma in repeaters reflected this group's overall behavioral pattern (Fig. 1f, right).

### Action history signals in parietal and motor cortex

Because the mapping between choice ("stronger" vs. "weaker") and motor action (left vs. right button press) varied between participants,

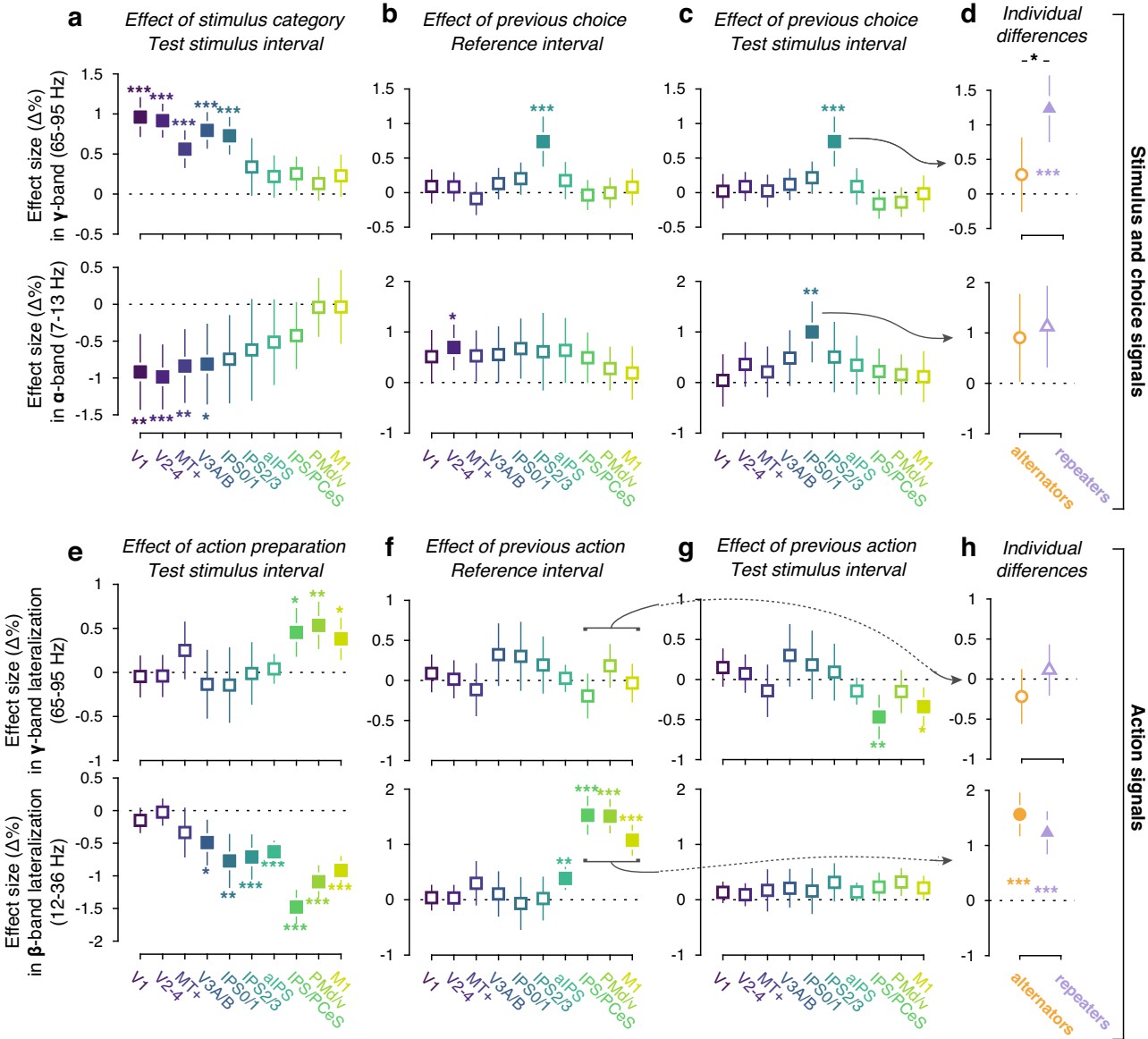

**Fig. 4 | Task-related and choice history signals across cortical areas.** Effect sizes from a general linear mixed model (see "Methods") estimating impact of stimulus category (**a**), choice (**b–d**), and action (**e–h**) on MEG power, in different regions and frequency bands. **a** Effect of current stimulus category on gamma (top) and alpha (bottom) power during test stimulus ($n = 60$). **b** As **a**, but for effect of previous choice in the reference interval. **c** As **b**, but for test stimulus interval. **d** Comparison of choice history signals between alternators ($n = 25$; orange) and repeaters ($n = 34$; purple). Group effect = 0.55328, CI [0.19342, 0.91314], $p = 0.00258$. **e** Effect of current action preparation on gamma (top) and beta (bottom) power during test stimulus ($n = 60$). **f** As **e**, but for effect of previous action during reference interval. **g** As **f**, but for the test interval. **h** Comparison of action history signals between alternators ($n = 25$; orange) and repeaters ($n = 34$; purple). Data are shown as average fixed effect size +/− 95% confidence intervals; *$0.01 < p < 0.05$ (no such effect present); **$0.001 < p < 0.01$; ***$p < 0.001$; filled markers, $p < 0.05$ (all $p$ values FDR corrected).

we could disentangle the described effects of choice history from effects of action history (i.e., the previous motor response) in our group analysis. Signatures of action preparation in beta-band lateralization (Fig. 3a) were present in multiple parietal and frontal cortical areas beyond the M1 hand area: IPS0-3, anterior intraparietal cortex (aIPS), the junction of intraparietal and postcentral sulci (IPS/PostCeS), and dorsal/ventral premotor cortex (PMd/v; Fig. 4e, bottom). By contrast, gamma lateralization was confined to IPS/PostCeS, PMd/v, and M1 (Fig. 4e, top). The effect of action history was present in M1, PMd/v, and IPS/PostCeS during reference (Fig. 4f, bottom), but less robustly during test (Fig. 4g, bottom). Action history signatures (beta-power lateralization pooled across IPS/PostCeS, PMd/v, and M1) did not differ between repeaters and alternators (Fig. 4h). The action history signal was only present after correct choices (Supplementary Fig. 3d, e). It did

not correlate with IPS2/3 gamma choice history signals (across-participant correlation: $r = 0.075$, $p = 0.5689$, $Bf_{10} = 0.1191$). None of the cortical history signals reflected choices or actions beyond one past trial (Supplementary Fig. 4e). Action history signals in motor cortex decayed from the start of the trial, no longer significant by the time the next trial's test stimulus is presented (Fig. 5c). Since observers were allowed to blink their eyes and make small movements during the inter-trial interval (thereby minimizing ocular and muscle artifacts during the trial), we could not track the emergence of choice or action history signals between trials.

In sum, we identified three neural signals that encoded different aspects of the previous choice: the perceptual decision and the motor act used to report that decision. Following "stronger" compared to "weaker" choices, both (i) gamma-band activity in IPS2/3 during test

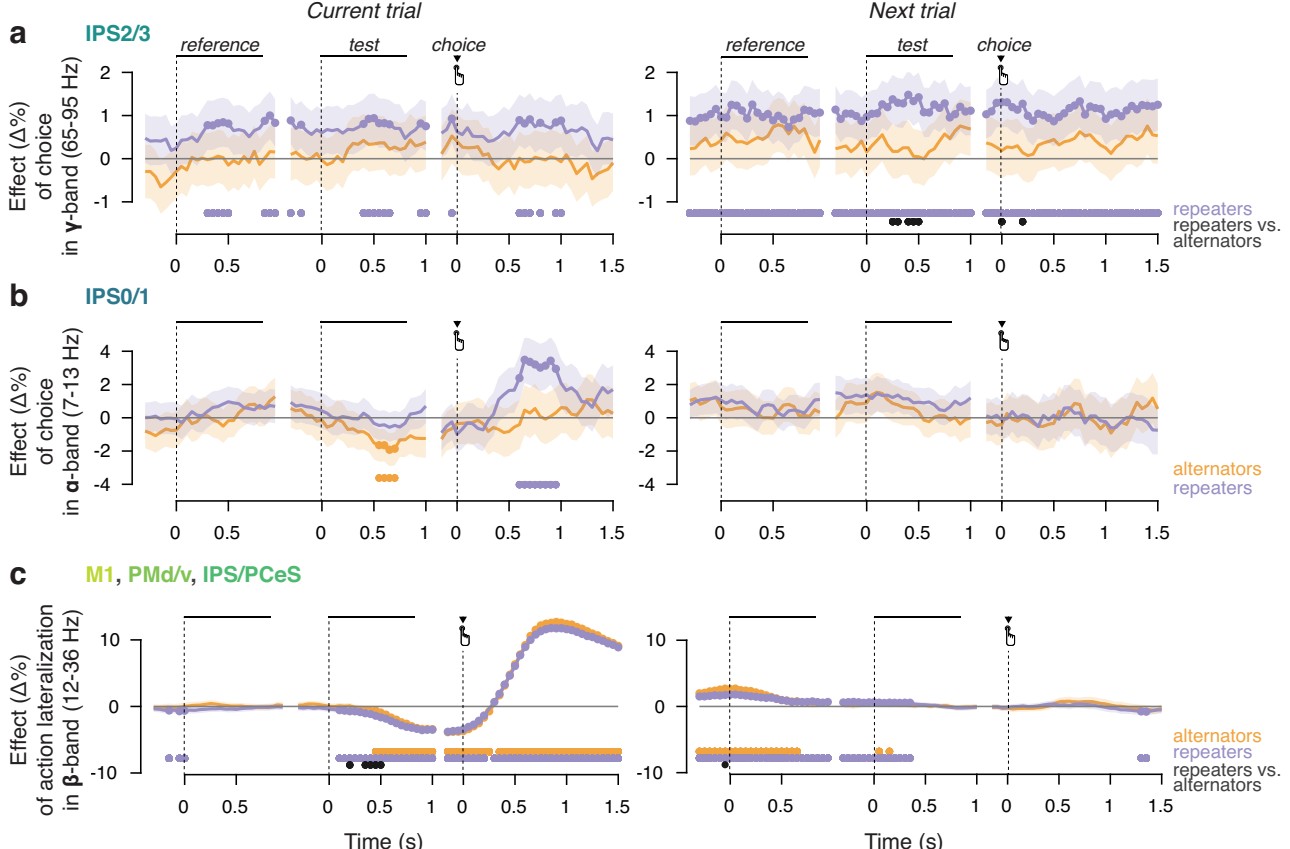

**Fig. 5 | Time course of history effects in cortical signals.** Time-resolved effect sizes from a general linear mixed model (see "Methods"), separately for both subgroups. **a** Effect of current and previous choice on IPS2/3 gamma-band activity. **b** Effect of current and previous choice on IPS0/1 alpha-band activity. **c** Effect of current and previous action on motor lateralization (pooled signal from M1, PMd/v, IPS/PCeS) in the beta-band. Lower markers indicate timepoints where the fixed effect is significantly different within each group, or between groups ($p < 0.05$, FDR-corrected). Data are shown as average fixed effect size +/− 95% confidence intervals.

and reference intervals, and (ii) alpha-band activity in IPS0/1 during the test stimulus interval, were enhanced. Additionally, (iii) beta-band activity in motor areas (IPS/PostCeS, PMd, and M1) was stronger contralateral than ipsilateral to the previous motor response, only during the reference interval. These three signals had dissociable anatomical, temporal and frequency profiles, and differed in their sensitivity to the outcome (correct or error) of previous choices. For brevity, we refer to these as IPS2/3-gamma, IPS0/1-alpha, and motor-beta (average of IPS/PostCeS, PMd, and M1), respectively. Note that the former two were extracted during the test stimulus interval, during which the decision was computed. The motor-beta signal was extracted from the reference interval, as its history modulation had vanished by the time of test stimulus viewing. Crucially, only the IPS2/3 gamma-band signal differed between subgroups of alternators and repeaters, suggesting it as a prime candidate for dominating behavioral choice history tendencies.

**Impact of cortical history signals on behavioral choice**
We then aimed to pinpoint the functional roles of these distinct neural history signals. We first ran a mediation analysis based on single-trial regressions (Fig. 6). In our model, the current choice was a categorical response variable (all regressions on that variable were logistic, see Methods), previous choice was a (categorical) regressor, and IPS2/3-gamma, IPS0/1 alpha, and motor-beta signals were included as candidate mediators of the impact of previous on current choice (Fig. 6a). Only the IPS2/3-gamma signal ($t(59) = 3.971$, $p = 0.0002$) but none of the other two signals (IPS0/1 alpha: $t(59) = 1.592$, $p = 0.1167$; motor beta: $t(59) = 1.178$, $p = 0.2434$) mediated (partially) the effect of previous choice on current choice (Fig. 6b). While significant for the

complete group, this effect was specifically expressed in repeaters, not alternators (Fig. 6). This group difference was most strongly driven by the effect of previous choices on IPS2/3-gamma (the a-path), rather than by the effect of this neural signal on the next choice (the b-path) (Supplementary Fig. 7). The mediating effect on choice of IPS2/3 gamma was present also when calculated selectively for previous choices that were correct, but not incorrect (Supplementary Fig. 8). The latter may reflect a lack of power due to the lower number of error trials, which is also suggested by the similar pattern for the direct effect (Supplementary Fig. 8). The direct path was also significant ($t(59) = 2.148$, $p = 0.0358$; Fig. 6b, right), indicating that the IPS2/3-gamma signal did not fully explain choice history biases ("partial mediation"). This is not surprising given that single-trial MEG signals are coarse population proxies of the cellular signals that drive behavior.

In our task, the parietal IPS2/3 gamma signal dominated overt choice history biases and its individual differences (differentiating between repetitive and alternating strategies), and mediated choice sequences. This was not the case for the motor beta lateralization, contradicting previous results: Pape and Siegel ref. 24 found that motor beta lateralization predicts choice alternation, in a motion discrimination task where perceptual choices and motor actions were decoupled on a trial-by-trial basis. Given this previous work, we wondered if the pre-stimulus motor beta signal had any effect on choice history behavior in our task. Specifically, this neural signal may have had a subtle influence on decision-making only visible in the distribution of reaction times, which, given the long overall decision times in our task, may not necessarily translate into biases in the final choice[7]. We next tested this idea by means of computational modeling of

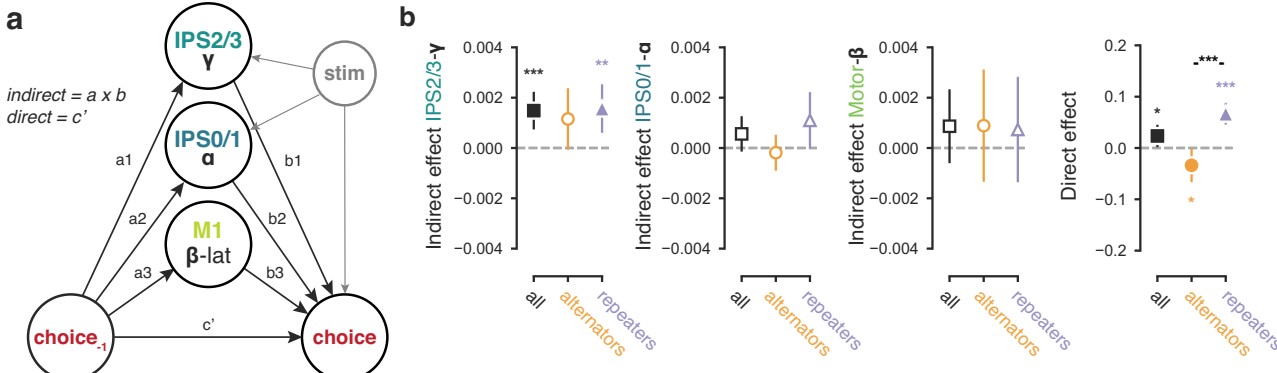

**Fig. 6 | Cortical IPS2/3-gamma signal mediates choice history bias. a** Schematic of mediation model. All effects pointing to the binary choice were computed using logistic regression. **b** Mediation parameter estimates for the complete group ($n = 60$) as well as both subgroups ($n = 25$ alternators in orange, $n = 34$ repeaters in purple). Left-right: indirect paths, quantifying mediation by neural signals. IPS2/3-gamma: all $t(59) = 3.971$, $p = 0.0002$; alternators: $t(24) = 1.948$, $p = 0.0632$; repeaters: $t(33) = 3.310$, $p = 0.0023$. IPS0/1-alpha: all $t(59) = 1.592$, $p = 0.1167$; alternators:

$t(24) = -0.543$, $p = 0.5923$; repeaters: $t(33) = 1.988$, $p = 0.0552$. Motor-beta: all $t(59) = 1.178$, $p = 0.2434$; alternators: $t(24) = 0.822$, $p = 0.4193$; repeaters: $t(33) = 0.713$, $p = 0.4807$. Right: direct path ($c'$) from previous to current choice: all $t(59) = 2.148$, $p = 0.0358$; $t(24) = -2.124$, $p = 0.0441$; repeaters: $t(33) = 6.415$, $p = 2.86e{-}007$. Data are shown as mean +/− 95% confidence intervals, statistics from a simple $t$ test against zero or between groups. *$0.01 < p < 0.05$; **$0.001 < p < 0.01$; ***$p < 0.001$; filled markers, $p < 0.05$.

reaction times and choices, using the different neural history signals as single-trial physiological regressors[42,43].

## Parietal and motor cortical signals play distinct roles in evidence accumulation

Current models of decision dynamics posit the temporal accumulation of sensory evidence, resulting in an internal decision variable that grows with time[44–47]. When this decision variable reaches one of two bounds, a choice is made and the motor response is initiated. Different history-dependent neural signals may be the source of distinct biases in evidence accumulation that we have previously uncovered through behavioral modeling[7]. One possibility is that biased activity states in motor circuitry add to accumulated evidence, without interacting with the accumulation process per se. Alternatively, biases in neural circuitry upstream from the evidence accumulator[38,48] may bias the evidence accumulation process directly. These scenarios can be disentangled behaviorally using accumulation-to-bound models, such as the widely used drift diffusion model (DDM). In this model, choice bias can arise from two different sources (Fig. 7a). On the one hand, an offset prior to the decision (and independent of the accumulation process) shifts the decision variable closer to one of the two decision bounds (*starting point bias*). On the other hand, the input to the evidence accumulator can be biased throughout decision formation, affecting the decision process just like a bias in the physical stimulus, and producing a stimulus-independent asymmetry in the rate of accumulation towards one versus the other bound (*drift bias*). These mechanisms can produce the same bias in choice fractions, but have distinct effects on the shape of reaction time distributions (Fig. 7a). Specifically, starting point biases most strongly drive choices that are made quickly, whereas drift biases accumulate over time and predict biased choices also when reaction times are slow[7,49,50].

We reasoned that the parietal signals (IPS2/3-gamma and/or IPS0/1-alpha) during test interval may bias evidence accumulation toward choice repetition, while the motor beta signal (average of IPS/PostCeS, PMd, and M1) at the start of the trial (reference interval) may bias the starting point towards choice alternation. The underlying rationale was that the parietal signals were present during decision formation, occurred in regions that were likely to be upstream from a putative evidence accumulator[51] and (for IPS0/1) encoded the decision-relevant sensory signal. The parietal signals also went in the direction of choice repetition (i.e., same sign as the effect of previous stimulus category). By contrast, the beta signal in action-related areas was expressed only

before decision formation, and pointed toward choice alternation (opposite sign as the effect of previous choices, Fig. 4f).

We fitted sequential sampling models to the neural and behavioral data to test these hypotheses and quantified the impact of trial-to-trial neural signals on drift bias and starting point (Methods). In line with previous work[7], the data were best captured by a nonlinearly collapsing bound (Supplementary Fig. 10a, "Methods"), and showed the expected strong effect of stimulus category on drift (Supplementary Fig. 10d). We replicated the main result from previously reported, standard DDM fits[7]. Specifically: (i) at the group level, previous choices had a negative effect on starting point (i.e., towards choice alternation) and a positive effect on drift (i.e., toward repetition; Supplementary Fig. 10e); (ii) individual differences in overt repetition behavior were better explained by the effect of choice history on drift bias, rather than starting point (Supplementary Fig. 10f). We then replaced the previous choice predictor with three single-trial neural signals, to assess if they predicted trial-to-trial variations in starting point or drift.

Parietal and motor signals mapped onto distinct components of evidence integration. IPS2/3-gamma predicted a positive modulation of drift bias ($p = 0.0426$) (i.e., in the direction of choice repetition), but had no effect on starting point ($p = 0.1484$; Fig. 7b). This effect was also present when using neural data from the reference interval (Supplementary Fig. 11a), and it was robust to removal of the impact of current stimulus category from the single-trial neural signals (by subtracting mean neural signal for each stimulus category; Supplementary Fig. 11b) and inclusion of a regressor for the previous choice (Supplementary Fig. 11c). In contrast, motor-beta during reference predicted a negative modulation of starting point ($p = 0.0448$) (i.e., in the direction of choice alternation), but no effect on drift bias ($p = 0.1543$; Fig. 7d), an effect that was neither significant during the delay interval, nor during the test interval (Supplementary Fig. 12). The residual IPS0/1-alpha signal did not predict either computational parameter (Fig. 7c). There was no significant interaction between the group (repeaters vs. alternators) and these neural regression patterns (Supplementary Fig. 13).

Previous behavioral modeling work has shown that adjustments of decision bounds play a key role in mediating the effects of outcome[52] and/or subjective confidence[53] from the previous trial on current decision formation. We, therefore, fitted further models to test if any of the history-dependent neural signals (IPS0/1-alpha, IPS2/3-gamma, motor-beta) or the previous choice per se modulated the bound height. There was no evidence for a trial-by-trial adjustment of decision bounds by previous choices (Supplementary Fig. 14) nor by history-dependent neural signals (Supplementary Fig. 15).

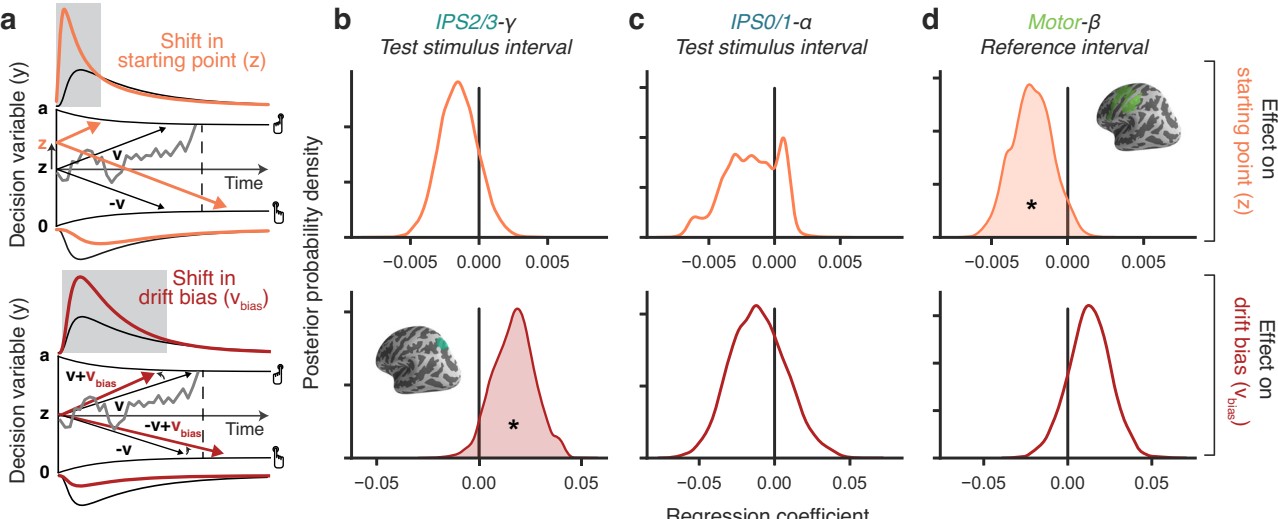

**Fig. 7 | Cortical signals predicting starting point and bias of evidence accumulation. a** Schematic of biasing mechanisms within DDM with collapsing bounds. Gray line, example single-trial trajectory of decision variable. Black lines, mean drift and RT distributions for unbiased decisions. Gray-shaded area: part of RT distributions for which biased parameter translates into overt choice bias. Orange lines: mean drift and RT distributions for a bias in starting point. Red lines: mean drift and RT distributions for a bias in drift. Adapted under a CC-BY license from: Urai et al.[7]. **b–d** Group-level posterior distributions over HDDMnn regression coefficients (see "Methods"), quantifying impact of neural signals (columns), on starting point (top) and drift bias (bottom). The model included nonlinearly collapsing bounds, non-decision time, overall (history-independent) starting point and drift, as well as a stimulus-dependent drift term. *$p < 0.05$.

## Discussion

Even in the face of random stimulus sequences, the tendency to systematically repeat or alternate previous choices is ubiquitous in decision-making[3,7,9–11]. Recent studies have begun to identify neural traces of choice history in brain regions implicated in perceptual decision-making[1,13,14,17,20,24,27,31], but these studies have commonly focused on one brain region at a time. Here, we combined human MEG recordings with behavioral modeling to dissect and compare multiple neural choice history signals that were distributed across the cortical sensory-motor pathway for an elementary visual decision and expressed in choice- or action-selective band-limited activity that persisted from one trial to the next. We focused on multiple visual field maps in occipital, temporal, and parietal cortex (Wang et al.[54]) and anterior parietal and frontal areas involved in action preparation[37,38,55]. To constrain the functional interpretation of the thus identified neural history signals, we compared them with established MEG signatures of sensory encoding and action preparation in the above regions.

Our approach uncovered a neural signature of choice repetition bias: persistent gamma-band activity in the intraparietal sulcus (higher-tier visual field maps IPS2/3). This parietal gamma-band signal encoded previous choices ("stronger" vs. "weaker" motion) in a sustained fashion from the previous choice up into the subsequent decision interval, forming an active bridge between successive decisions. The signal was strongly related to idiosyncratic features of sequential behavior: it was only expressed in observers with a tendency to repeat their choices. Critically, in these repeaters, it mediated choice repetition bias. Replicating a previous study[24], we also found that the sustained lateralization of beta-band activity in motor cortical areas (IPS/PostCeS, PMd/v, and M1) encoded the motor act (left vs. right button press) used to report the perceptual judgment on the previous trial. However, this action history signal had a negligible effect of mediating choice history biases and did not differ between observers with repetitive vs. alternating behavior. Using neurally informed accumulation-to-bound models, we found that the motor-beta signal biased the starting point of evidence accumulation toward choice alternation, while the parietal-gamma signal biased the rate of evidence accumulation toward choice repetition. This starting point effect of the motor-beta signal extends a previous demonstration of this signal's

involvement in choice alternation[24] by linking it to a specific computational parameter. Taken together, our results indicate that persistent activity in intraparietal cortex reflect action-independent choice history signals, which shape choice repetition behavior. Posterior parietal cortex may thus integrate and sustain decision-relevant information not only within[45], but also across trials, providing a bridge between sensory responses and longer-lasting beliefs about the structure of the environment.

Neural correlates of decision biases have commonly been studied in the context of explicit experimental manipulations of e.g., stimulus probabilities, rewards or single-trial cues. By contrast, the biases we describe here are intrinsic and highly idiosyncratic. They arise despite verbal instructions to focus only on sensory stimuli, cannot fully be eliminated with extensive training[3,13], and are strongly idiosyncratic[7]. The ultimate source of these individual differences remains a target for speculation. Agents may differ in their representation of the stability of the environment, yielding distinct different history biases[56]. The IPS2/3 gamma signal identified here was specific to observers who tended to repeat their choices, pointing to a potential role in implementing an idiosyncratic assumption of environmental stability.

Our findings corroborate recent demonstrations of choice history signals in the posterior parietal cortex of mice[17,18,31] and rats[19,20]. History biases in rodents may specifically depend on PPC neurons that project to the striatum, rather than motor cortex[31], highlighting how specific populations of neurons may be involved in distinct decision-making computations. Because of the diminished sensitivity of our MEG recordings[57] for subcortical regions, we did not analyze subcortical areas in the present study. The correlative nature of our recordings precludes strong inferences about the causal role of human parietal history signals[28], but optogenetic inactivation of rodent posterior parietal cortex reduces history dependencies in behavior[18,20]. Notably, this was observed only for inactivation prior to decision formation, and parietal inactivation during the next stimulus left choice history biases unaffected[18,20,58]. The latter observation, obtained in rodents and different tasks, seems to be at odds with our finding that the human IPS2/3-gamma signal during stimulus viewing was a significant mediator of choice history biases. This apparent inconsistency points to the need for combined inactivation and recording studies and more direct

cross-species comparisons, in order to better understand the specific role of parietal cortical dynamics in shaping choice sequences.

Our behavioral task was designed so that we could exploit a previously established motion coherence-dependent MEG signals[34] as selective markers for the encoding of the near-threshold task-relevant sensory signals (difference in motion strength; Fig. 2). Further, different from two-interval discrimination tasks used in the somatosensory and auditory domains[20,59,60] we kept the reference stimulus constant across trials. Participants could thus use a stable category boundary for "stronger" vs. "weaker" (learned by practice before the MEG recordings), instead of working memory representation that was newly formed on each trial. We reasoned that this might render choice history effects in our task less dependent on working memory of the reference. This assumption was supported by the persistent nature of choice history signals observed in IPS2/3, which were largely uninterrupted by the reference (Fig. 5, top), as well as by the lack of behavioral effect of trial-by-trial variations in persistent activity during the delay interval, over and above the long-lasting signal inherited from the previous trial (Supplementary Fig. 9). Taken together, these features of the task design enabled us to precisely relate the spatial and spectral properties of choice history signals to the encoding of decision evidence, and show that attractive history biases are also present in tasks where working memory is not needed, extending recent rodent work[20].

A biasing effect of a sustained neural signal on the rate of evidence accumulation ("drift bias") may reflect a biased encoding of the sensory information[61]. Our results argue against this possibility: such an effect would have to be expressed in a region and frequency band that also encodes the stimulus itself. IPS2/3 was the first visual field map in the hierarchy where this was not the case. Another possibility is a non-sensory bias signal that feeds into the accumulator together with the stimulus information, adding a history-dependent bias to the accumulation process. The IPS2/3-gamma is consistent with this scheme, specifically for subjects with a tendency to repeat. Such a scheme would also translate into the observed stimulus-independent drift bias and resulting bias in choice behavior.

Previous behavioral work has pointed to a link between response preparation and the starting point of evidence accumulation[25,49,50]. Action-specific beta-power lateralization in the cortical motor system pushes the motor state away from the most recent response[24,29]. Combining physiology with single-trial behavioral modeling enabled us to show that this motor beta signal specifically biased the starting point of evidence accumulation toward response alternation. This beta signal is also found in pure motor tasks that do not entail decision-making, and it likely reflects idling in motor circuits[41] that prevents action repetition. In our task, this signal did not have a detectable effect on overt choice sequences. This is largely in line with previous work: repetitive choice history biases persist even with variable stimulus response mapping[1,6,24,26], and inactivation of PPC-M2 projection neurons does not abolish choice history bias in mice[18]. In our experiment, the choice-response mapping was fixed within individuals, different from the variable mapping used by Pape and Siegel[24]. This may have caused the motor history signal to decay as participants formed the next decision, reducing the impact of the beta rebound on subsequent choices. The dominant effect of accumulation biases (toward repetition) over starting point biases (toward alternation) on choices may arise from their temporal dynamics: the slower the decision, the more drift biases dominate over starting-point biases in their effect on choices[7,49,50]. So, we could only identify the effect of motor beta-rebound on starting point by jointly fitting choices and response time distributions.

Our modeling approaches could be extended in several instructive ways. First, the DDM is a simplified model of the dynamics of evidence integration[44]. For example, it is unlikely that integration is non-leaky and that drift biases are constant. We previously established

that even in such cases, choice history most strongly affects the accumulation bias, rather than the starting point of evidence integration[7]. Future work could combine time-resolved sensory inputs with fitting of more complex decision-models[7,47,62], to help refine the within-trial time course of choice history biases across cortical areas. Second, although history effects on behavior were long-lasting, we here only used the previous trial's choice as a proxy. Various reinforcement-learning models have been used to account for choice history bias[1], its dependence on confidence[22,63] and its response to environmental statistics[56,64]. Another important avenue would be to relate model-derived single-trial estimates of behavioral history bias[15] and/or drift[65] to the rate of decision-related cortical build-up activity[35,37,38,66].

In conclusion, our results show that choice history is an important source of trial-to-trial variability in cortical dynamics, which in turn biases subsequent decision computations and choice behavior. These results contribute to our understanding of how decision processes arise from a rich interplay of sensory information and contextual factors across the cortical hierarchy.

## Methods

### Participants

Sixty-four participants (aged 19–35 years, 43 women and 21 men) participated in the study after screening for psychiatric, neurological or medical conditions. All participants had normal or corrected to normal vision, were non-smokers, and gave their informed consent before the start of the study. The experiment was approved by the ethical review board of the University Medical Center Hamburg-Eppendorf (reference PV4648). Before each experimental session, participants were administered a pill containing donepezil (5 mg Aricept®), atomoxetine (40 mg Strattera®) or placebo (double-blind cross-over design). These pharmacological manipulations did not affect behavioral choice history biases[7], and were therefore not incorporated in the analyses presented here.

Three participants did not complete all sessions of the experiment and were thus excluded. After rejecting trials with excessive recording artifacts (see below), we discarded one additional participant with fewer than 100 trials per session remaining. In total, 60 participants were included in the analysis.

### Behavioral task

Participants were asked to judge if the coherence of a random-dot motion stimulus in a so-called test interval was stronger or weaker than a preceding reference stimulus, which was shown afresh on each trial at 70% coherence (Fig. 1a). A red "bulls-eye" fixation target of 0.6° diameter[67] was present in the center of the screen throughout the experiment. Each trial started with a baseline interval of 500–1000 ms of randomly moving dots (0% coherence). A beep (50 ms, 440 Hz) indicated the onset of the reference stimulus (70% coherence) that was shown for for 750 ms. The reference was followed by a variable (300–700 ms) delay (0% coherence). An identical beep indicated the onset of the test stimulus, whose motion coherence deviation from the 70% reference toward stronger or weaker coherence. The deviation was individually titrated prior to the main experiment (see below). A counterbalancing scheme ensured that each stimulus category (weaker or stronger) was followed by the same or the other category equally often[68]. During both non-zero coherence (i.e., reference and test) intervals, dots moved in one of the four diagonal directions, counterbalanced across and constant within participants. After the offset of the test stimulus, observers reported their judgment ("stronger" vs. "weaker") by pressing a button with their left or right index finger (response deadline of 3 s). The hands used to report "weaker" and "stronger" judgments were counterbalanced across participants. Feedback (correct/incorrect) was then indicated by a tone of 150 ms

(880 or 200 Hz, feedback-tone mapping counterbalanced between participants).

## Stimuli

Random dot kinematograms were presented in a central annulus (outer radius 14°, inner radius 2°) around fixation. The annulus was defined by a field of dots with a density of 1.7 dots/degrees². Dots were white with 100% contrast from the black background and 0.2° in diameter. Signal dots were randomly selected on each frame, moved with 11.5°/second in one of four diagonal directions and had a limited "lifetime" of four consecutive frames, after which they were replotted in a random location. Signal dots that left the annulus wrapped around and reappeared on the other side. Noise dots were assigned a random location within the annulus on each frame, resulting in "random position" noise with a "different" rule[69]. Moreover, three independent motion sequences were interleaved on subsequent frames to prevent tracking of individual signal dots[70].

## Experimental procedure

Before their first MEG session, participants received instructions and then did one behavioral session to determine their 70% correct threshold for the main experiment. First, 600 trials with test stimuli containing 1.25, 2.5, 5, 10, 20, and 30% coherence difference (from the 70% coherence reference) were randomly interleaved. The inter-stimulus interval was 1 s, and participants took a short break after each set of 125 trials. They did not receive feedback. Stimuli were presented on an LCD screen at 1920 × 1080 resolution and 60 Hz refresh rate, 60 cm away from the participants' eyes. To determine each individual's psychometric threshold, we fit a cumulative Weibull as a function of absolute coherence difference $c$, defined as

$$\psi(c) = \delta + (1 - \delta - \gamma)\left(1 - e^{-\left(\frac{c}{\alpha}\right)^{\beta}}\right) \qquad (1)$$

where $\delta$ is the guess rate (chance performance), $\gamma$ is the lapse rate, and $\alpha$ and $\beta$ are the threshold and slope of the psychometric Weibull function, respectively[71]. While keeping the guess rate $\delta$ bound at 50% correct, we fit the parameters $\alpha$, $\beta$, and $\gamma$ using a maximum likelihood procedure implemented by minimizing the logarithm of the likelihood function. This was done using a Nelder–Mead simplex optimization algorithm as implemented in Matlab's *fminsearch* function. The individual threshold was taken as the stimulus difficulty corresponding to a 70% correct fit of the cumulative Weibull.

Second, participants performed another 100 trials using a 2-up 1-down staircase procedure. This procedure accounted for any learning effects or strategy adjustments during thresholding. The coherence difference between the two stimuli started at their 70% correct threshold as obtained from the Weibull fit. It was increased by 0.1% coherence on making an error, and decreased by 0.1% on giving two consecutive correct answers. Thresholds from this staircase ranged from 3.3 to 13.4% (mean 6.9%) motion coherence difference.

Participants then performed the task at their individual motion threshold for a total of 1.200 trials during two MEG sessions (600 trials each, 13–30 days apart). Between these two MEG sessions, they performed three practice sessions (1500 trials each, on separate days) outside the MEG. In the behavioral practice sessions, we presented feedback immediately after the participants' response. An ISI of 1 s was observed before continuing to the next trial. Participants completed training on 4500 trials, over 3 separate sessions, between the two MEG recordings. The training data are not used in our current analyses.

## Quantification of choice history weights

We quantified individual choice history strategies by fitting an extended logistic regression model, as described in[3,6,12]. This approach extends the psychometric function with a history-dependent bias term $\delta_{\text{hist}}(\boldsymbol{h_t})$,

reflecting was a linear combination of previous stimuli and choices

$$P(r_t = 1 | \widetilde{s}_t, \boldsymbol{h_t}) = \gamma + (1 - \gamma - \lambda) g(\delta(\boldsymbol{h_t}) + \alpha(\widetilde{s}_t)) \qquad (2)$$

where $\lambda$ and $\gamma$ were the probabilities of stimulus-independent errors ("lapses"), $\widetilde{s}_t$ was the signed stimulus intensity, $g(x) = 1/(1 + e^{-x})$ was the logistic function, $\alpha$ was perceptual sensitivity. The bias term $\delta(\boldsymbol{h_t})$ was the sum of the overall bias $\delta'$ and the history-dependent bias $\delta_{\text{hist}}(\boldsymbol{h_t}) = \sum_{k=1}^{K} \omega_k h_{kt}$, where $\omega_k$ were the weights assigned to each previous stimulus or choice. We set $K = 7$, and included both previous stimuli and previous choices. Each set of seven past trials was convolved three exponentially decaying basis functions[3]. Positive history weights $\omega_k$ then indicated a tendency to repeat the previous choice, or to make a choice that matched the previous stimulus. Negative weights described a tendency to alternate the corresponding history feature. All parameters were fit using an expectation maximization algorithm[3].

## Behavioral streak analysis

Following ref. [1], we analyzed multi-trial build-up of choice history bias by extracting choice repetition sequences of increasing length. This was first done separately for the two choice identities ("stronger" vs. "weaker"). We then quantified "repeating" bias as the shift in SDT criterion on the subsequent trial $t + 1$, taking into account that trial's stimulus category. To quantify a repeating bias irrespective of the choice identity, we afterwards sign-flipped those criterion values that followed a "weaker" choice. This resulted in a value that, at sequence length 1, reflected each individual's repetition probability (as in Fig. 1c). We then extended this to longer sequences of repeated choices, where the last trial was either again a repetition or the first trial to deviate from the repetition streak. To additionally test for a potential reset of choice history bias after errors[8], we repeated the analysis after sub-selecting for the last trial being either correct or incorrect (Supplementary Fig. 1c).

## MEG data acquisition

MEG was recorded using a 275-channel CTF system in a shielded room. Horizontal and vertical EOG, bipolar ECG, and an electrode at location POz (about 4 cm above the inion) were recorded simultaneously. All signals were low-pass filtered online (cut-off: 300 Hz) and recorded with a sampling rate of 1200 Hz. To minimize the displacement of the subject's head with respect to the MEG sensors, we used online head-localization[72] to show the head position to the subject inside the MEG chamber before each block. Participants were then asked to move themselves back into their original position, correcting slow drift of their head position during the experiment. Between the two recording days, the original head position from day one was used as a template for day two.

Stimuli were projected into the MEG chamber using a beamer with a resolution of 1024 × 768 pixels and a refresh rate of 60 Hz. The screen was positioned 65 cm away from participants' eyes. Horizontal and vertical gaze position and pupil diameter were recorded at 1000 Hz using an MEG-compatible EyeLink 1000 on a long-range mount (SR Research) at 60 cm from the subject's eye. The eye tracker was calibrated before each block of training.

## Structural MRI

Structural T1-weighted magnetization prepared gradient-echo images (TR = 2300 ms, TE = 2.98 ms, FoV = 256 mm, 1 mm slice thickness, TI = 1100 ms, 9° flip angle) with $1 \times 1 \times 1$ mm³ voxel resolution were obtained on a 3 T Siemens Magnetom Trio MRI scanner (Siemens Medical Systems, Erlangen, Germany). Fiducials (nasion, left and right intra-aural point) were marked on the MRI.

## Preprocessing of MEG data

MEG data were analyzed in Matlab using the Fieldtrip Toolbox[73] and custom scripts. MEG data were first resampled to 400 Hz and epoched into single trials from baseline to 2 s after feedback. We removed trials where the displacement of the head was more than 6 mm from the first trial of each recording. Trials with SQUID jumps were detected by fitting a line to each single-trial log-transformed Fourier spectrum, and rejecting trials where the intercept was detected as an outlier based on Grubb's test. To remove the effect of line noise on the data, we computed the cross-spectrum of the data at 50 Hz, resulting in a complex matrix of size $n$-by-$n$, where $n$ was the number of channels. We applied singular value decomposition to this cross-spectrum and took the first eigenvector (corresponding to the largest singular value) as the spatial topography reflecting line noise. The two-dimensional space spanned by the real and imaginary parts of this eigenvector was then projected out of the data, effectively suppressing any signal that co-varied with activity at 50 Hz. Line noise around 50, 100 and 150 Hz was then removed by a band-stop filter, and each trial was demeaned and detrended.

We also removed trials containing low-frequency artifacts from cars passing by the scanner building, muscle activity, eye blinks, or saccades. These were detected using FieldTrip's automated artifact rejection routines, with rejected thresholds determined per recording session by visual inspection.

## Spectral analysis of MEG data

We computed time-frequency representations for each of the four epochs of interests, analyzing low and high frequency ranges separately. For the low frequencies (3–35 Hz in steps of 1 Hz), we used a Hanning window with a length of 400 ms in steps of 50 ms and a frequency smoothing of 2.5 Hz. For high frequencies (36–120 Hz in steps of 2 Hz), we used the multitaper technique with five discrete proloid slepian tapers[74] a window length of 400 ms in steps of 50 ms, and 8 Hz frequency smoothing. For sensor-level analyses, the data from each axial gradiometer were decomposed into two planar gradients before estimating power and combined afterwards, to simplify the topographical representation of task-related power modulations.

The time–frequency representations were converted into units of percent power change from the pre-trial baseline. The baseline power was computed as the across-trial mean from −300 to −200 ms before reference onset, separately for each sensor and frequency. The resulting time-frequency representations were further averaged across trials and sensors of interest.

## MEG source reconstruction

We estimated power modulation values for a set of cortical regions of interest (ROIs, see below) based on source-reconstructed voxel time courses from a sliding window DICS beamformer[75,76]. A source model with 4 mm resolution was created from each individual's MRI, and warped to the Colin27 brain[77] using a nonlinear transformation for group averaging.

Within the alpha (8–12 Hz), beta (12–36 Hz), and high-gamma (65–95 Hz) bands, we computed a common filter based on the cross-spectral density matrix estimated from the first 2 s of each trial, starting from the start of the baseline time window. For each grid point in the brain, we then applied the beamformer (i.e., spatial field) in a sliding window of 250 ms, with steps of 50 ms. The resulting source estimates of band-limited power were again converted into units of percent power change from the trial-average baseline, as described above. Rare outliers with values larger than 500 were removed. All further analyses and modeling were applied to the resulting power modulation values.

## Selection of MEG sensors exhibiting sensory or motor signals

To select sensors for the unbiased quantification of visual responses, we computed power modulation in the gamma-range (65–95 Hz) from 250 to 750 ms after test stimulus onset, and contrasted trials with stronger vs. weaker visual motion. We then selected the 20 most active sensors at the group level, in the first and second session separately. This procedure yielded stable sensor selection across sessions (Fig. 2b, inset). 18 sensors were selected in both sessions (circles), two were selected only in the first session (downward-pointing triangles) and one was selected only in the second session (upward-pointing triangle).

Similarly, for sensors corresponding to response preparation, we contrasted trials in which the left vs. the right hand was used to respond. We computed power in the beta range (12–36 Hz) in the 500 ms before button press[35], and used the same split-half approach to define the 20 most active sensors for the contrast left vs. right, as well as the 20 most active sensors the for opposite contrast, to extract left and right motor regions (Fig. 3a, inset).

For each session, we then extracted single-trial values from the sensors defined on the other session. Note that sensor selection was only used for visualizing TFRs, not for the beamformed signals that we later use for statistics and modeling (see below).

## Definition of cortical ROIs

Following previous work[37,38], we defined a set of ROIs spanning the visuo-motor cortical pathway from the sensory (V1) to the motor (M1) periphery. The exact delineation of ROIs was based on anatomical atlases from previous fMRI work, specifically: (i) retinotopically organized visual cortical field maps[54] along the dorsal visual pathway up to IPS3; (ii) three regions exhibiting hand movement-specific lateralization of cortical activity: aIPS, IPS/PostCeS and the hand sub-region of M1[55]; and (iii) a dorsal/ventral premotor cluster of regions from a whole-cortex atlas[78]. We grouped visual cortical field maps with a shared foveal representation into clusters[79] (Table 1), thus increasing the spatial distance between ROI centers and minimizing the risk of signal leakage (due to limited filter resolution or volume conduction). We selected all grid points located within each grouped ROI, and averaged their band-limited power signals. PySurfer (https://pysurfer.github.io/) was used to visualize each ROI on an inflated cortical surface.

## Time windows and signals of interest

We selected two time-windows for in-depth statistical modeling: (i) the test interval during which the decision was formed (0–750 ms after test stimulus onset); and (ii) for pre-decision state, the reference interval (0–750 ms after reference stimulus onset). For each trial, we averaged the power modulation values across all time bins within these two time-windows, and used the resulting scalar values for further analyses.

The general linear modeling (see below) was applied to each ROI individually. For subsequent mediation and drift diffusion modeling, we further focused on three signals of interest: IPS2/3 gamma during the test stimulus (reflecting choice history), IPS0/1 alpha during the test stimulus (reflecting choice history after error trials), and motor beta (pooled across IPS/PostCeS, M1 and PMd/v) during the reference (reflecting action history). Choice-action mapping was counterbalanced across participants. We flipped motor lateralization signals for half of the participants, so that the lateralization was always computed with respect to the hand reporting "stronger" choices.

The choice history-dependent IPS0/1-alpha signal was superimposed by a spatially non-specific suppression of alpha power following error feedback, which was shared by all cortical ROIs but not related to specific choice history (Supplementary Fig. 3). We averaged this global signal across all visual field map ROIs except IPS0/1, and removed it from the IPS0/1-alpha power modulation values via linear projection[80–82]. We used this residual IPS0/1-alpha, unconfounded by the global signal, for all subsequent behavioral modeling.

## Statistical assessment of power modulation values

At the sensor-level (see definition of sensors of interest above) we used cluster-based permutation testing across the group of participants[83] to find clusters, for which trial-averaged power modulation values differed across the group of participants for a given contrast of interest. For the assessment of full time-frequency representations of power modulation, clusters were two-dimensional, defined across time and frequency.

We used general linear mixed effects models (GLMEs, using Matlab's *fitglme*) to quantify the effect of choice history on single-trial power modulation values across all source-level ROIs, frequency ranges and the above-defined time windows. The model included a random intercept for each participant:

$$n \sim 1 + s + c_{-1} + (1|p) \tag{3}$$

where $n$ was the single-trial neural data (at a specific time window, region of interest and frequency band), $s$ indicated the stimulus category $[-1, 1]$, $c_{-1}$ indicated the behavioral choice $[-1, 1]$ on the preceding trial, and $p$ was a participant identifier.

We also tested if the effect of previous choices differed between participants with choice repetition probabilities larger ("repeaters") or smaller ("alternators") than 0.5. We first estimated effect sizes and confidence intervals separately for these two groups, using the model above. This was repeated after randomly sub-selecting 25 "repeaters," to match the number of participants between the two groups. We also fitted the group interaction term explicitly:

$$n \sim 1 + s + g*c_{-1} + (1|p) \tag{4}$$

where $g$ was coded as $[-1, 1]$, reflecting if a participant showed overall alternation vs. repetition behavior. All estimated effects (with confidence intervals and FDR-adjusted $p$ values) are available as a Supplementary File, see "Data availability."

## Statistics

We tested for the effect of sequence length (1–5) and sequence end (a matching or non-matching final trial) on behavioral and neural repetition biases. Within each subgroups this was done using pingouin's repeated-measures analysis of variance (ANOVA)[84]. To additionally test for the between-subjects factor of subgroup, we used the mixed repeated-measures ANOVA in JASP[85].

Throughout, error bars show 95% confidence intervals. These were obtained through bootstrapping (behavioral and mediation figures) or from the GLME model fits (neural signals). For the neural GLMEs, p-values were corrected for multiple comparison across all ROIs, frequency bands and time windows by controlling the False Discovery Rate, the fraction of false positives, at 0.05[86]. All statistical tests reported are two-sided.

## Mediation modeling

To estimate the causal effect of trial-by-trial neural signals on choice behavior, we performed a mediation analysis using the *lavaan* package[87]. We fit the following regression equations

$$\begin{aligned} \boldsymbol{\beta} &\sim a_1\boldsymbol{c}_{-1} + s_1\boldsymbol{s} \\ \boldsymbol{\alpha} &\sim a_2\boldsymbol{c}_{-1} + s_2\boldsymbol{s} \\ \boldsymbol{\gamma} &\sim a_3\boldsymbol{c}_{-1} \\ \boldsymbol{c} &\sim b_1\boldsymbol{\beta} + b_2\boldsymbol{\alpha} + b_3\boldsymbol{\gamma} + c'\boldsymbol{c}_{-1} + s_0\boldsymbol{s} \end{aligned} \tag{5}$$

where $\gamma$ was the single-trial IPS2/3 gamma, $\alpha$ was the single-trial IPS/01 alpha residual, $\beta$ was the single-trial pooled motor-beta, $c$ was a vector of choices, and $s$ was a vector with stimuli (−1 "weak" vs. 1 "strong"). We

then defined our effects of interest as follows:

$$\begin{aligned} \text{indirect:} &= ab \\ \text{direct:} &= c' \\ \text{total:} &= ab + c' \end{aligned} \tag{6}$$

We fit the model separately for each participant using a WLSMV estimator, and then computed group-level statistics across the standardized individual coefficients. Data were analyzed with *pandas* and *pingouin*[84] and visualized with *seaborn*[88].

## Drift diffusion modeling

To fit a set of Hierarchical Drift Diffusion models with trial-by-trial MEG regressors, we used the HDDMnn package[42,43]. In each model, we fit a stimulus-dependent drift rate $v$, starting point $z$, boundary separation $a$, and nondecision time $t$. First, we assessed if our data was best described by a static, linearly collapsing, or collapsing bound. The latter, best-fitting bound collapse was described by a Weibull function

$$b(t; a, \alpha, \beta) = a * e^{\left(-\frac{t^\alpha}{\beta}\right)} \tag{7}$$

We then let both the starting point and drift bias of the DDM depend on single-trial neural data:

$$v \sim 1 + s + n_\gamma + n_\alpha + n_\beta \tag{8}$$

$$z \sim n_\gamma + n_\alpha + n_\beta \tag{9}$$

where $v$ was the drift, $z$ the starting point, $s$ the stimulus category $[-1,1]$, and $n_x$ were vectors of single-trial neural signals of interest (IPS2/3 gamma, IPS0/1 alpha, and motor beta lateralization), each $z$-scored within participant. The model used "stimulus coding," which fits response time distributions for "stronger" vs. "weaker" choices (rather than for correct vs. incorrect choices) in order to capture a selective choice bias.

For each model, we ran a Markov Chain Monte Carlo procedure with 10,000 samples, of which a tenth was discarded as burn-in. Five percent of trials were assumed to be outliers in the fitting procedure. Statistics were computed on the group-level posteriors.

## Reporting summary

Further information on research design is available in the Nature Research Reporting Summary linked to this article.

## Data availability

The processed behavioral and ROI data generated in this study have been deposited in the OSF database under accession code https://osf.io/v3r52/. The raw MEG data are under restricted access (due to the consent form used at time of data collection), and are available upon request from AEU.

## Code availability

All codes used to run the task, process data, and generate figures are available at https://doi.org/10.5281/zenodo.6949711.

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

## Acknowledgements

We thank Jan Willem de Gee for discussions on DDM analyses and Anke Braun for comments on the manuscript. Christiane Reißmann, Karin Deazle, Samara Green and Lina Zakarauskaite helped with participant recruitment and data collection. Roger Zimmermann provided technical assistance during MEG recordings and Guido Nolte provided valuable advice and code for removing line noise artifacts. Andreas Engel pro-vided funding support for MEG recordings and pharmacology. This work was supported by the German Academic Exchange Service (DAAD), German National Academy of Sciences Leopoldina, and International Brain Research Organization (to AEU) and the Deutsche Forschungsge-

meinschaft (DFG, German Research Foundation), SFB 936—178316478/Z3, DO 1240/2-1, DO 1240/2-2, DO 1240/3-1, and DO 1240/4-1 (to T.H.D.). Analyses were carried out on the Dutch national e-infrastructure with the support of SURF Cooperative, and on the Academic Leiden Interdisciplinary Cluster Environment (ALICE) provided by Leiden University.

## Author contributions

Conceptualization: A.E.U., T.H.D.; methodology, software, investigation, formal analysis, data curation, visualization: A.E.U.; writing: A.E.U., T.H.D.; supervision: T.H.D.; funding acquisition: A.E.U., T.H.D.; project management: A.E.U., T.H.D.

## Funding

## Competing interests

The authors declare no competing interests.
