## [Peer Review File · Nature Communications]

Persistent Activity in Human Parietal Cortex Mediates Perceptual Choice Repetition BiasREVIEWER COMMENTS

Reviewer #1 (Remarks to the Author):

The study can be considered a follow up from the authors' previous work, in which they show that biases in the drift rate of evidence accumulation (as opposed to starting point) can explain behavioral findings from a variety of perceptual paradigms (Urai et al., 2019). Here, they surpass some of the limitations associated with behavioral modelling, combining it with brain data, a very powerful way of inference (Turner et al., 2016).

Authors do a good job reviewing previous findings. They refer to a set of studies which show history effects on choice in either the motor or parietal cortices, and point out that none of them have systematically compared the influence of both or related them to evidence accumulation (EA), which, to my knowledge, is true. By using behavioral modeling and MEG, the authors disentangle two components present in choice history effects. Although the paper is very well-written and convincing, there are some issues in interpreting the results more precisely.

Major Points

1. The authors just analyzed one-trial-to-one trial dynamics. One possible comment could be if sequential effects extend further back from the last trial (two trials back, three trials back, ...), and whether the tendency to repeat the same choice is enhanced the more the choice has already been repeated. This could be done both by analyzing behavioral and neural data. Also, this may be easier to detect if only using the subset of participants the authors refer to as "repeaters" (participants with a greater tendency to repeat their choices).

2. The authors might need to comment on the issues regarding the nature of choice history dependency. Fischer & Whitney (2014, Nat Neurosc) claim that perception is attracted towards the previous choice. Fritsche et al. (2017, Curr Biol) reply that attractive history effects are decisional (and more specifically based on drifts in working memory), while the perceptual history effects are repulsive. The related results were presented in the behavioral experiments of Akaishi et al. (Neuron), which demonstrated that the choice history dependency is neither sensory nor motor, but decisional. These issues are relevant for the current study because some of the components, especially in the ones in the motor cortex, can be interpreted as motor but not decisional.

Minor Points

3. In the previous study done by Akaishi et al. (Neuron, 2014), the researchers identified the different neural structures from the current study. In the previous paper, the researchers were interested in history effects that can be long-lasting (spanning over several trials). The researchers applied an RL-like model to explain how people use internally and externally-generated signals to learn about the value of choices, and to update them from trial to trial. The authors in the current study are not interested in long-lasting effects, and their focus is just on the previous trial. The authors simply want to disentangle the different last-trial effects on the current trial. Since the focus is just on the previous trial, they do not employ learning models, and they use a DDM instead. Such modelling allows the authors to discover which aspect of evidence accumulation each component biases. These different foci between the previous studies and the current study might be worth mentioning.

4. The authors of the current manuscript come up with a paradigm that makes working memory not needed for the task. In their words: "Because the strength of the reference stimulus was fixed across trials, the task did not require comparing the test against a short-term memory representation of the reference stimulus (as is the case in widely used tasks with trial-to-trial variations of the reference stimulus, e.g. (Machens et al., 2005; Akrami et al., 2018)). Rather, participants could solve our task by using a stable category boundary (in our case for 'stronger' vs. 'weaker') learned during practice before the MEG recordings (see Methods)". But nothing more is said about it. I feel the authors miss a valuable opportunity to use the discussion to 1) mention again the advantages of their paradigm, 2) claim that attractive history effects are not only based on working memory, as they are present even when working memory is not involved.

5. The authors used a single model of evidence accumulation (EA). The interpretation of the behavioral and neural data depends on the underlying assumption of the chosen model. Utility of the interpretation of EA modelling requires accepting a set of non-unanimously agreed upon assumptions regarding the proposed algorithm (that choice evidence is sampled and integrated until reaching a predefined threshold), as well as neural implementation, such as functional role of individual neurons in the EA process (Barack and Krakauer, 2021). Much speculation is present in the literature whether sampling and accumulation to threshold metaphor is adequate to explain choice mechanisms. Other alternatives, such as urgency gating (Cisek et al., 2009, Thura et al., 2017), quantum walk (Kvam, Busemeyer & Pleskac, 2021), or stochastic transitions (Miller & Katz, 2010) have been proposed. Additionally, the family of sequential sampling models is quite large, and contains models that vary in assumptions (Evans & Wagemakers, 2019). Broad literature on this topic suggests that parameter interpretation can change significantly depending on the model used (e.g. Goldfarb et al., 2014). Because of this variety of the formulations of the EA process, the authors might wish to explain the reason for the choice of the model and make it explicit about the underlying assumptions.

6. On two occasions, the authors of the current paper defend that an effect is not present, but the effect seems to be marginal. The first case is on page 5, when saying that there was no main effect of the stimulus category on choice history effects within IPS2/3 ($p = 0.05$). The second time this happens is at the very end of page 9, when defending that the motor-beta signal did not modulate the drift rate of evidence accumulation ($p = 0.10$). In the presence of these marginal results, the authors could discuss whether they expect those effects not to be significant, even if power was increased, or whether there may be a solid trend there.

7. There are quite a lot of typos across the manuscript and the supplementary material: “behavioral evidence indicates points to” (Page 2, second paragraph), “refence” (Page 3, first paragraph), “more than expected from change” (Figure S12, caption), “systematic choice repetition was prevalent than” (Figure S12, caption). These are just four of the few that I’ve seen around.

8. Authors should also correct the mention they make of their figures. I suspect there was a recent rearrangement of the orientation of the different panels within the figures, and then the text was not updated. It is common, for instance, that authors bring the attention to a right panel, when they really mean the bottom panel. This generates quite a lot of confusion.

Reviewer #2 (Remarks to the Author):

In their work, Urai and Donner present a characterization of the relation between cortical rhythms and choice history biases in the domain of visual perception in humans. More specifically, the authors base their analyses on MEG signals, DDMs, and other statistical analyses to find and track which specific cortical oscillations relate to choice repetition biases relative to the choices made in the previous trial. The authors find that Gamma-band activity in the parietal cortex tracked previous choices and biased evidence accumulation toward choice repetition. Conversely, Beta activity in the motor cortex promoted choice alternation (relative to the previous trial). The results that the authors present are interesting, as they provide a dissociation between repetition/alternation biases and neural oscillations with its corresponding cortical areas involved in such processes (although it should be mentioned that the finding of response alternation guided by motor cortex is not new, as this was already characterized in previous work by Pape and Siegel, Nature Communications 2016). Having said this, there are various aspects that will be important to clarify to make sure that the claims and conclusions that the authors make are warranted by the set of analyses they present, as currently some of them appear somehow overstated. I expand on my comments below

1) One of my main criticisms is related to the order of the presentation of the behavioral results and the expectation created from the way the Introduction and Results are currently outlined. I find it a bit odd that the presentation of behavioral results comes only at the end of the Results section, without any mention to them early on. The Results section starts right away with the presentation of the MEG results. My suggestion is that the behavioral results are presented first to be able to understand what the characteristics of behaviour at the trial-by-trial and population levels are, that are intended to be related to neural dynamics. It is well taken that there is quite some interindividual variability in choice repetition bias, with no clear significant trend for repetition or alternation in their human population sample. I suggest that the authors are upfront by mentioning this interindividual variability, and therefore, that the goal is to study what may influence choice repetition on a trial-by-trial basis.

2) The second critical issue is related to the expectations that the Title and various passages of the Introduction generate. If I understood correctly, the analyses that the authors carry out early in the presentation of the Results are mostly related to *choice repetition* bias. There are no other aspects of “choice history” biases that are thoroughly characterized alongside choice repetition biases. This is interesting, because in many of the work cited in the Introduction, and also in work conducted the authors and their closed collaborators, other key aspects that make part of choice history biases are often jointly considered. For instance, one such factor commonly studied is the effect of the interaction between choices and reward (or correctness) in the previous trials which determines the “win-stay lose-switch” phenomenon, critically after controlling for the main effect of choice repetition. Why was not this also considered in this study? It is well possible that some of the variability explained in their neural regressions is partially related to win-stay lose-switch. My suggestion is that the authors incorporate this key aspect of history bias in both their neural and behavioral regressions. That is, these models should incorporate both factors simultaneously in their neural GLMs and DDMs.

Having said this, there is nothing wrong with the strategy of solely focusing on *choice repetition* biases in their manuscript, but then, for instance, the title should clearly reflect this aspect, and should not read as general as currently the title claims the study is. That is, the last part of the title should read “... Computation of Choice *Repetition* Biases” (it currently reads “... Computation of Choice *History* Biases”, which is not warranted by the analyses reported in this study). The same holds for various concluding statements in the Results and Discussion sections.

3) Related to the previous point (2), the study focuses on *perceptual decisions* and both the title and the abstract should reflect this aspect. I suggest that it reads something like “... Computation of Choice *Repetition* Biases in perceptual decision making”. It is unclear whether the findings that the authors present here also hold for decisions beyond the perceptual domain. Similar general conclusions should be avoided in the rest of the text.

4) The authors make use of DDM single-trial regressions to understand what latent variables of the choice process is influenced by neural signals in each cortical structure, which is great. However, it is possible that the cortical/latent-variable “dissociation” results they obtained are biased by the different latencies they used to fit the DDM single-trial regressions where the two brain areas are treated as completely independent. Not sure if this is what the authors did, but one way to alleviate this concern is to include both cortical structures in the same single-trial model and let the signals of the two cortical structures compete for explanatory variance in the same model. A second issue is related to the fact that decision bound was not included in the regression. It is well possible that history biases also affect decision bounds who trial by trial adjustment has been previously shown to play a key role in changes of mind. I strongly suggest to also include bound as a latent factor to be related to neural signals in the DDM regressions. Finally, all models should also include the effect of previous stimulus (as done in their behavioral model) to control for potential biases induced by spillovers of sensory evidence from previous trials.

5) The authors find that only alternators were influenced by Motor-beta. The first question is whether only alternators had an influence on starting point bias (z) in the DDM but not in the drift, which would be a direct prediction that would further validate the effectiveness of the DDM results. Second, if only repeaters were influenced by both parietal-gamma and motor-beta, mechanistically speaking, which of the two cortical signatures more strongly influenced choice repetition bias such that they turned into repeaters? As mentioned by the authors, one would expect that one cortical signature had more influence than the other otherwise the repetition bias effect would have cancelled out due to the opposing bias influence that the cortical signatures induce. This would help clarifying what neural signature exerts stronger influence on repetition biases

6) Minor: In the Introduction the authors criticize the lack of causal brain-bias relations in previous work giving the initial impression that they want to solve it, but at the end of the same paragraph they go back to a correlative method as core component of their study. I would suggest rephrasing this part.

7) Minor: In the first part of the Results the authors write the following: “Because the strength of the reference stimulus was fixed across trials, the task did not require comparing the test against a short-term memory representation of the reference stimulus...”. I am not sure this is entirely correct. By design, the participants still must hold a reference stimulus in mind to be compared with the test stimulus. If the intention was to avoid a working-memory component, why the paradigm was not designed such that it included only one stimulus such as discriminating motion direction (also possible with motion strength)? Please clarify.

8) Minor: I suggest to include Figure S2 in the main text as it provides relevant and rich information about the dynamics of key factors of repetition biases in the neural dynamics throughout the trial.

Reviewer #3 (Remarks to the Author):

In this paper, Urai & Donner use human MEG to investigate oscillatory power during perceptual choices, with the goal of disentangling neural signals related to choice and motor history, and how any biases impact on the (computational) decision process. They find oscillatory signals relating to choice and motor action, and differential roles in the evidence accumulation process.

While these are generally interesting results, I had some reservations.

1. If a proximal goal is simply to disentangle whether previous motor action or previous decisions decision bias future choice, I am unsure why that would require neural-level data. One could simply use variable (within participants) response mappings such as to determine whether people repeat their choice or their motor action? Are there no previous studies that have done this?
2. If both choice and motor action bias the evidence accumulation process, how can it be explained that only one of them biases the ultimate decision on the next trial?
3. In the results, it was often not clear to me what statistical test a statement was based on, and how each result accounted for the number of simultaneous tests involved (e.g. sensors, time and frequency windows, regions of interest). Given the multitude of reported results (and of perhaps equally plausible analyses), it might be useful to point out how their number is accounted for.
4. The authors sometimes use FDR correction; unlike FWE this implies that 5% of reported results are expected to be false positives; this should be noted.
5. The analysis has many degrees of freedom. Just to mention a few: the definition of frequency bands, of time windows for sensor selection and number of selected sensors, the definition of ROIs etc. How can the authors ensure that their statistical results are not due to particular settings of the variables?
6. Some statements about the absence of effects are based on non-significance in a NHST; this interpretation is not supported by the test (particularly not if the p-value is slightly above .05: "The

choice history effect in this area, correspondingly, did not interact with the effect of current stimulus category", p5 and other examples).

7. What is the rationale of reporting confidence intervals? Is this a journal requirement? It might be useful to point out their value (see e.g. (Morey et al., 2016) for why they are or can be misleading).

Morey RD, Hoekstra R, Rouder JN, Lee MD, Wagenmakers EJ (2016) The fallacy of placing confidence in confidence intervals. *Psychon Bull Rev* 23:103-123.

NCOMMS-21-40349: Response to reviewers

Dear editors and reviewers,

We thank the reviewers for their constructive and thoughtful evaluation of our manuscript. We have addressed each of the reviewers' concerns, by performing a large number of additional analyses, reordering results, and by substantially revising the entire manuscript.

Specifically, we have:

- reorganized the Results section so as to start with behavior and emphasize individual differences throughout. We also significantly changed all other figures from Figure 2 onwards so as to fully expose the individual differences in the identified neural history signals.
- included two completely new main figures, one showing longer-lasting choice history patterns in behavior, and one showing the within-trial time courses of these signals. We added eight additional supplementary figures.
- improved our drift diffusion model fits to include nonlinearly collapsing bounds (supported by formal model comparison) as well as joint fits of all three identified neural history simultaneously.

We believe that these major revisions have considerably strengthened the conclusions we can draw from the results. In what follows, we first provide a detailed point-by point reply to each of the reviewers' comments (printed in *blue italics*).

We hope that the reviewers will now find our manuscript suitable for publication in *Nature Communications*.

Sincerely,

Anne Urai and Tobias Donner

Point-by-point reply to reviewer #1

The study can be considered a follow up from the authors' previous work, in which they show that biases in the drift rate of evidence accumulation (as opposed to starting point) can explain behavioral findings from a variety of perceptual paradigms (Urai et al., 2019). Here, they surpass some of the limitations associated with behavioral modelling, combining it with brain data, a very powerful way of inference (Turner et al., 2016). Authors do a good job reviewing previous findings. They refer to a set of studies which show history effects on choice in either the motor or parietal cortices, and point out that none of them have systematically compared the influence of both or related them to evidence accumulation (EA), which, to my knowledge, is true. By using behavioral modeling and MEG, the authors disentangle two components present in choice history effects. Although the paper is very well-written and convincing, there are some issues in interpreting the results more precisely.

Thank you for your positive assessment of our work, and for your constructive suggestions.

Major Points

1. The authors just analyzed one-trial-to-one trial dynamics. One possible comment could be if sequential effects extend further back from the last trial (two trials back, three trials back, ...), and whether the tendency to repeat the same choice is enhanced the more the choice has already been repeated. This could be done both by analyzing behavioral and neural data. Also, this may be easier to detect if only using the subset of participants the authors refer to as "repeaters" (participants with a greater tendency to repeat their choices).

In the previous version of the manuscript, we indeed limited our analyses to one-trial-back history effects. This approach was motivated by the previous observation that the inter-individual variability in history biases seems particularly pronounced for the effects of immediately preceding trial (Urai et al., 2019; their Figure 5c).

We now more fully explore longer-lasting dynamics, both at the behavioral (Figure 1d,f and Figure S1) and neural level (Figure S4). We use two complementary methods to do so, in each case separating between groups of repeaters and alternators as you suggested.

First, we fit expanded logistic regression models (Fründ et al., 2014; Braun et al., 2018) to quantify the impact of up to 7 previous choices and stimuli (Figure S1a,b). As in previous reports, we observe that the effect of previous choices is strongest for lag 1, and differs from zero until around lag 3 or 4 (beyond lag 5, there is no discernible effect). These choice history weights, but not the weights for previous stimuli, strongly differentiate between groups of alternators and repeaters. Beyond showing the effects of longer-lasting previous trial event, this analysis also provides evidence that the main effect of interest (individual differences in behavior) are related to previous choice, not stimuli (see also your point 2).

Second, we quantify multi-trial 'streaks' to assess the build-up of higher-order history patterns across specific sequences of choices, as in (Akaishi et al., 2014), their figure 1E (Figure 1f, Figure S4). Importantly, we find that repetition effects in both behavior and IPS gamma-band power increase after longer sequences of repetitions, specifically in repeaters.

Thank you for this suggestion. We believe these across-trial patterns are a good complement to the within-trial dynamics we later explore in the DDM and that they further strengthen our conclusions regarding group differences.

2. The authors might need to comment on the issues regarding the nature of choice history dependency. Fischer & Whitney (2014, Nat Neurosc) claim that perception is attracted towards the previous choice. Fritsche et al. (2017, Curr Biol) reply that attractive history effects are decisional (and more specifically based on drifts in working memory), while the perceptual history effects are repulsive. The related results were presented in the behavioral experiments of Akaishi et al. (Neuron), which demonstrated that the choice history dependency is neither sensory nor motor, but decisional. These issues are relevant for the current study because some of the components, especially in the ones in the motor cortex, can be interpreted as motor but not decisional.

We agree that this is an important point, and now elaborate on this in the Introduction. We also cite behavioral models (Braun et al., 2018; Zhang and Alais, 2020; Feigin et al., 2021), who have all addressed this question through behavioral manipulations and/or modelling.

As to the first point (sensory vs. decisional effects) we have now specifically tested the effects of previous choices and previous stimuli to disentangle them:

- First, in Figure S1b, the weights of previous stimuli at lag 1 are not different from zero, and do not differ between alternators and repeaters (contrast this with the strong effects, and subgroup differences, in the weights of previous choices).
- Second, in Figure S5 and S6, we show the within-trial regression weights (using a sliding window GLM) of both previous choices, and previous stimuli. This shows that previous stimuli have no significant effect on any of the neural signals in parietal or motor cortex.

Both analyses strengthen our interpretation that we are mostly dealing with effects of previous choices. This is especially likely in our task design, because any perceptual aftereffects would be expected to be overwritten by the baseline and reference random dot stimuli.

As to the second point (decisional vs. motor), we fully agree with the conclusion by (Akaishi et al., 2014) that pure motor effects play a negligible role in behavioral choice history biases. In fact, our own work has taken a different modelling approach (Braun et al., 2018) and arrived at the same conclusion.

Yet, (Pape and Siegel, 2016) reported that motor lateralization were strongly predictive of response alternation in different versions of random dot motion tasks (with known and unknown stimulus-response mappings at the time of evidence presentation). In our work, we aimed to directly compare these different accounts, and to find the differential effects of parietal decisional vs. motor signals. We now more clearly highlight this in the Introduction and in the Results section (p. 11). Also, Figure 7 sheds some light on this issue through DDMs (motor beta affect starting point, with only a negligible effect on overall choices), which we believe helps clarify these contrasting ideas in the literature.

Minor Points

3. In the previous study done by Akaishi et al. (Neuron, 2014), the researchers identified the different neural structures from the current study. In the previous paper, the researchers were interested in history effects that can be long-lasting (spanning over several trials). The researchers applied an RL-like model to explain how people use internally and externally-generated signals to learn about the value of choices, and to update them from trial to trial. The authors in the current study are not interested in long-lasting effects, and their focus is just on the previous trial. The authors simply want to disentangle the different last-trial effects on the current trial. Since the focus is just on the previous trial, they do not employ learning models, and they use a DDM instead. Such modelling allows the authors to discover

which aspect of evidence accumulation each component biases. These different foci between the previous studies and the current study might be worth mentioning.

We agree and have now, in the Discussion, explicitly clarified the difference in the focus of the work from (Akaishi et al., 2014) as well as (Lak et al., 2020; Mendonça et al., 2020) who use reinforcement-learning type models to account for choice history biases.

4. The authors of the current manuscript come up with a paradigm that makes working memory not needed for the task. In their words: “Because the strength of the reference stimulus was fixed across trials, the task did not require comparing the test against a short-term memory representation of the reference stimulus (as is the case in widely used tasks with trial-to-trial variations of the reference stimulus, e.g. (Machens et al., 2005; Akrami et al., 2018)). Rather, participants could solve our task by using a stable category boundary (in our case for ‘stronger’ vs. ‘weaker’) learned during practice before the MEG recordings (see Methods)”. But nothing more is said about it. I feel the authors miss a valuable opportunity to use the discussion to 1) mention again the advantages of their paradigm, 2) claim that attractive history effects are not only based on working memory, as they are present even when working memory is not involved.

Thank you for this great suggestion. We have now added a paragraph to the Discussion, which elaborates on this point. We agree with your assessment, but also take into account one of Reviewer 2’s minor points (their Point 2), which questions the absence of working memory demands in our task. So, we have toned this claim down a bit.

We also ran an additional mediation analysis to test for any effects of the delay interval following the reference stimulus on choice, with the following rationale: if trial-by-trial variations in a working memory representation of the reference stimulus would play a significant role in the decision process, then the IPS2/3 gamma-band signal *during the delay interval* should mediate choice history bias, over and above the IPS2/3 gamma-band signal during the test stimulus. To isolate trial-by-trial variations in a putative activity component encoding reference working memory, we used the difference between delay activity and activity measured during the baseline interval on the same trial. We found no significant effect of this proxy of a putative reference working memory signal on choice, either across all participants or in subgroups of repeaters and alternators (Figure S9).

6. On two occasions, the authors of the current paper defend that an effect is not present, but the effect seems to be marginal. The first case is on page 5, when saying that there was no main effect of the stimulus category on choice history effects within IPS2/3 ($p = 0.05$). The second time this happens is at the very end of page 9, when defending that the motor-beta signal did not modulate the drift rate of evidence accumulation ($p = 0.10$). In the presence of these marginal results, the authors could discuss whether they expect those effects not to be significant, even if power was increased, or whether there may be a solid trend there.

We have now toned down those statements.

7. There are quite a lot of typos across the manuscript and the supplementary material: “behavioral evidence indicates points to” (Page 2, second paragraph), “refence” (Page 3, first paragraph), “more than expected from change” (Figure S12, caption), “systematic choice repetition was prevalent than” (Figure S12, caption). These are just four of the few that I’ve seen around.

Thank you very much for pointing us to these typos. We have now thoroughly checked the whole manuscript and hope it is free of typos.

8. Authors should also correct the mention they make of their figures. I suspect there was a recent rearrangement of the orientation of the different panels within the figures, and then the text was not updated. It is common, for instance, that authors bring the attention to a right panel, when they really mean the bottom panel. This generates quite a lot of confusion.

Thank you for spotting this. All references to the figures should now be accurate.
Thank you very much again for your insightful and constructive comments.

Point-by-point reply to reviewer #2

In their work, Urai and Donner present a characterization of the relation between cortical rhythms and choice history biases in the domain of visual perception in humans. More specifically, the authors base their analyses on MEG signals, DDMs, and other statistical analyses to find and track which specific cortical oscillations relate to choice repetition biases relative to the choices made in the previous trial. The authors find that Gamma-band activity in the parietal cortex tracked previous choices and biased evidence accumulation toward choice repetition. Conversely, Beta activity in the motor cortex promoted choice alternation (relative to the previous trial). The results that the authors present are interesting, as they provide a dissociation between repetition/alternation biases and neural oscillations with its corresponding cortical areas involved in such processes (although it should be mentioned that the finding of response alternation guided by motor cortex is not new, as this was already characterized in previous work by Pape and Siegel, Nature Communications 2016).

We are delighted that you find our results interesting. We did not intend to suggest that the beta-band related response alternation is novel. Instead, we largely used this previously established ‘action history signal’ as a reference, against which we compared the functional properties and computational signatures of the key novel finding of our current work: the parietal history signal expressed in the gamma-band. We now make this more explicit in the revised Introduction, highlight the (Pape and Siegel, 2016) findings, and also clarify the main gap in the literature that our current work aimed to address: untangling the relative contribution of multiple co-existing neural history signal, expressed in distinct brain regions.

We would like to highlight that we also extended the beta-band choice alternation effect identified in the paper by Pape and Siegel in an important way: pinpointing its effects on the evidence accumulation process (assessed through single-trial diffusion modelling), again in relation to the parietal gamma-band signal. This revealed a clear dissociation between both neural history signals.

Having said this, there are various aspects that will be important to clarify to make sure that the claims and conclusions that the authors make are warranted by the set of analyses they present, as currently some of them appear somehow overstated. I expand on my comments below.

We have performed substantial textual revisions and several major new analyses to address your points.

1) One of my main criticisms is related to the order of the presentation of the behavioral results and the expectation created from the way the Introduction and Results are currently outlined. I find it a bit odd that the presentation of behavioral results comes only at the end of the Results section, without any mention to them early on. The Results section starts right away with the presentation of the MEG results. My suggestion is that the behavioral results are presented first to be able to understand what the characteristics of behaviour at the trial-by-trial and population levels are, that are intended to be related to neural dynamics. It is well taken that there is quite some interindividual variability in choice repetition bias, with no clear significant trend for repetition or alternation in their human population sample. I suggest that the authors are upfront by mentioning this interindividual variability, and therefore, that the goal is to study what may influence choice repetition on a trial-by-trial basis.

Thank you for this suggestion. We had discussed this issue among ourselves while composing the original version of the manuscript, and ultimately converged on the order you saw initially. Following your suggestion, we have now reassessed this and decided to re-arrange the entire Results section accordingly, as summarized below.

We now start the manuscript with Figure 1 (and associated Figure S1) that quantifies the individual differences in behavioral choice history patterns and defines the separation between repeaters and alternators, to be used throughout the subsequent MEG analyses. Here, we now go beyond single-trial history effects, incorporating analyses of multi-trial streaks.

Then, rather than presenting the individual differences in neural signals as a standalone final figure (old figure 7), we now include panels on the group split of effects across all main MEG figures (Figures 4-6 and associated supplementary figures).

We believe that these changes have substantially strengthened the manuscript, putting the focus on the individual differences in choice history bias throughout. We thank you for this valuable suggestion.

*2) The second critical issue is related to the expectations that the Title and various passages of the Introduction generate. If I understood correctly, the analyses that the authors carry out early in the presentation of the Results are mostly related to *choice repetition* bias. There are no other aspects of "choice history" biases that are thoroughly characterized alongside choice repetition biases. This is interesting, because in many of the work cited in the Introduction, and also in work conducted the authors and their closed collaborators, other key aspects that make part of choice history biases are often jointly considered. For instance, one such factor commonly studied is the effect of the interaction between choices and reward (or correctness) in the previous trials which determines the "win-stay lose-switch" phenomenon, critically after controlling for the main effect of choice repetition. Why was not this also considered in this study? It is well possible that some of the variability explained in their neural regressions is partially related to win-stay lose-switch. My suggestion is that the authors incorporate this key aspect of history bias in both their neural and behavioral regressions. That is, these models should incorporate both factors simultaneously in their neural GLMs and DDMs.*

*Having said this, there is nothing wrong with the strategy of solely focusing on *choice repetition* biases in their manuscript, but then, for instance, the title should clearly reflect this aspect, and should not read as general as currently the title claims the study is. That is, the last part of the title should read "... Computation of Choice *Repetition* Biases" (it currently reads "... Computation of Choice *History* Biases", which is not warranted by the analyses reported in this study). The same holds for various concluding statements in the Results and Discussion sections.*

In the first paragraph of the Introduction, we now describe these different names for very similar phenomena. We also define clearly our usage of the term 'choice history bias' throughout the manuscript, which we prefer to remain consistent with our own recent work on the topic (Braun et al., 2018; Urai et al., 2019). Because the key new history signal identified here (parietal gamma) is specifically expressed in people with a tendency to repeat (not alternate) their choices, we have now focussed the title on choice repetition bias.

We also note that individual differences in effect of previous choices dominate those of previous stimuli (Figure S1), justifying our focus here.

As for the title, we now follow your suggestion and changed it to "Persistent Activity in Human Parietal Cortex Mediates Perceptual Choice Repetition Bias".

*3) Related to the previous point (2), the study focuses on *perceptual decisions* and both the title and the abstract should reflect this aspect. I suggest that it reads something like "... Computation of Choice *Repetition* Biases in perceptual decision making". It is unclear whether the findings that the authors*

present here also hold for decisions beyond the perceptual domain. Similar general conclusions should be avoided in the rest of the text.

We agree. We have now specified by adding 'perceptual' throughout.

4) The authors make use of DDM single-trial regressions to understand what latent variables of the choice process is influenced by neural signals in each cortical structure, which is great. However, it is possible that the cortical/latent-variable "dissociation" results they obtained are biased by the different latencies they used to fit the DDM single-trial regressions where the two brain areas are treated as completely independent. Not sure if this is what the authors did, but one way to alleviate this concern is to include both cortical structures in the same single-trial model and let the signals of the two cortical structures compete for explanatory variance in the same model.

Indeed, we previously fitted each neural regressor separately. We have now incorporated your suggestion, and included all three neural regressors into this model. As expected from the negligible correlation between single-trial IPS2/3 gamma and motor beta signals (group average $r = 0.002$, t -value across subjects = 0.073, $p = 0.9419$), we find the same results even with this joint model. We think this result is much stronger than the previously reported one. Thank you for this suggestion.

As to your point regarding the neural signals' latencies: we chose these based on the time window in which each (choice/action) history signal was strongest. This is now more clearly pointed out on page 10 in the manuscript. Two control analyses more specifically address the latency of our DDM effects. The IPS2/3 gamma-band effect on drift bias is also significant in the reference interval (Figure S11a), in line with this signal's sustained nature throughout the trial (Figure S5a). In contrast, the motor beta effect on starting point is no longer significant beyond the reference interval (Figure S12b,c), again in line with the time course of these action history signals (Figure 3, Figure 5c).

A second issue is related to the fact that decision bound was not included in the regression. It is well possible that history biases also affect decision bounds whose trial by trial adjustment has been previously shown to play a key role in changes of mind. I strongly suggest to also include bound as a latent factor to be related to neural signals in the DDM regressions.

We have addressed this point in the following ways. First, we tested if our assumption of a stationary bound may not be the best fit to these data. Indeed, this revealed that a model with a nonlinearly collapsing bound (modelled as a Weibull function) fit the data best (Figure S10a). We now adopt these models throughout. Second, we ran a model where each neural signal predicted bound height. This yielded no effect, as shown in the new Figure S15. Third, we tested if previous choice had any impact on bound height, again yielding no effect as shown in the new Figure S14. Taken together, these new results indicate that (i) within-trial bound dynamics are important to explain our data, but (ii) there is no evidence for a trial-by-trial adjustment of decision bounds dependent on choice history and/or history-dependent neural signals.

Please note that we are not suggesting that there are no systematic trial-by-trial bound adjustments *at all* – based on previous findings, we expect bounds to be modulated by the outcome feedback (Purcell and Kiani, 2016) and/or subjective confidence (Desender et al., 2019) from the previous trial. But these effects are orthogonal to the selective biasing effects by the identity of the previous choice, which is the focus of our current study.

Finally, we would like to point that we also tried fitting a model, in which each neural signature could affect starting, point, drift bias, and bound height. Fits of this (complex) model overall confirm our

previous conclusions (Figure R1): no effect of neural signals on bound height; remaining negative effect of motor beta on starting point, and remaining positive effect of parietal gamma on drift. The latter effect was only marginally significant ($p = 0.0636$), possibly due to the increased complexity and trade-off between parameters. We do not feel sufficiently confident in these last fits to report them in the paper. The large number of parameters included made the fitting difficult, and it did not seem to converge in several occasions (showing divergent posterior distributions when re-running the models). Given the complete absence of any detectable effect of neural signals and previous choice on bound height in the simple models reported above, we think there is no need to include such hard-to-interpret results.

Figure R1. HDDMnn models including bound height. As Figure 7, but with each neural signal affecting trial-to-trial fluctuations in starting point (top), drift bias (center) or bound height (bottom).

5) The authors find that only alternators were influenced by Motor-beta. The first question is whether only alternators had an influence on starting point bias (z) in the DDM but not in the drift, which would be a direct prediction that would further validate the effectiveness of the DDM results.

Both alternators and repeaters had a significant effect of previous actions on motor beta lateralization (new Figure 4h). We did not observe a group difference in this effect, nor in the mediation effect of motor beta onto choice sequences (new Figure 6b). It thus seems that the effects of action history over motor cortex, at least in our dataset, do not reflect individual differences in behavior.

As you suggested, we also tested for a group difference in the DDMs, running the ‘full’ model (with all three neural parameters and a nonlinearly collapsing bound) that now included an additional interaction term with the group (Figure S13). Here, we found no significant group effect, suggesting that the factor driving individual differences is the effect of choice history on parietal gamma-band power rather than the effect of this neural signal on subsequent decisions. This distinction is further confirmed by separately investigating the a-path and the b-path in our mediation analysis (Figure S7), which we now discuss in the Results section.

Second, if only repeaters were influenced by both parietal-gamma and motor-beta, mechanistically speaking, which of the two cortical signatures more strongly influenced choice repetition bias such that they turned into repeaters? As mentioned by the authors, one would expect that one cortical signature

had more influence than the other, otherwise the repetition bias effect would have cancelled out due to the opposing bias influence that the cortical signatures induce. This would help clarifying what neural signature exerts stronger influence on repetition biases.

We have now included the group split (alternators vs. repeaters) in our mediation analysis (new Figure 6) in order to address which of the two cortical signatures more strongly influenced individual sequential behavior. This shows that *only* the parietal-gamma signature mediates history bias, and it does so only in repeaters, not in alternators. By contrast, motor-beta does not show any significant mediation effect, neither in the complete sample of subjects, nor in any individual sub-group. This analysis pinpoints the parietal-gamma signal as a likely (and unique) driver of idiosyncratic choice repetition bias.

6) Minor: In the Introduction the authors criticize the lack of causal brain-bias relations in previous work giving the initial impression that they want to solve it, but at the end of the same paragraph they go back to a correlative method as core component of their study. I would suggest rephrasing this part.

Thank you for pointing us to this issue, which was indeed confusing. We have now revised the Introduction accordingly.

7) Minor: In the first part of the Results the authors write the following: "Because the strength of the reference stimulus was fixed across trials, the task did not require comparing the test against a short-term memory representation of the reference stimulus...". I am not sure this is entirely correct. By design, the participants still must hold a reference stimulus in mind to be compared with the test stimulus. If the intention was to avoid a working-memory component, why the paradigm was not designed such that it included only one stimulus such as discriminating motion direction (also possible with motion strength)? Please clarify.

We understand your point, but maintain our statement that our task did not *require* comparing the test against a short-term memory representation of the reference stimulus: because the reference was constant across trials, subjects could form a stable representation (stored in a longer-term form of memory, e.g., involving synaptic plasticity) of that category boundary that does not need to be updated on every single trial. This was made more likely by the behavioral practice subjects received prior to MEG recordings (see Methods).

This aspect of our task design is a critical difference to two-interval discrimination tasks used previously in the somatosensory and auditory domains (Romo and Salinas, 2003; Machens et al., 2005; Akrami et al., 2018), in which the reference stimuli varied from trial to trial and thus needs to be stored in working memory. This design has given rise to a distinct pattern of choice history effects in rodents, also linked to parietal cortex (Akrami et al., 2018). Of course, our design feature does not *exclude* the possibility that subjects do use a working memory representation of the reference; it only ensures that working memory is not strictly *necessary* to solve the task.

We have now clarified the overall reasoning underlying our choice of task design in a dedicated the discussion paragraph in the revised manuscript.

To strengthen this point further, we have now also included results from additional mediation analysis to control for any effects of persistent activity encoding the reference stimulus during the delay: if trial-by-trial variations in a working memory representation of the reference stimulus would play a significant role in the decision process, then the IPS2/3 gamma-band signal *during delay* should mediate choice history bias, over and above the IPS2/3 gamma-band signal during test. To isolate trial-by-trial variations in a putative activity component specific for reference working memory, we computed the

difference between delay activity and activity measured during the baseline interval on the same trial. We found no significant effect of this proxy of a putative reference working memory signal on choice, either across all participants or in subgroups of repeaters and alternators (Figure S9).

Finally, as for the question of why we showed the reference on each at all: this was the first two-interval forced choice task that we have used in our lab (and the second experiment of this kind). We were not sure if absence of repeated presentations of a reference stimulus would deteriorate performance. By now, we have performed versions of a similar task (contrast discrimination) (Wilming et al., 2020) with and without reference presentation, and there seems little difference in overall performance, in line with our assumption that most subjects tend not to use a working memory representation of the reference for the decision process in this type of task.

8) Minor: I suggest to include Figure S2 in the main text as it provides relevant and rich information about the dynamics of key factors of repetition biases in the neural dynamics throughout the trial.

We agree that the time courses are instructive and have now moved these panels showing the effects of current/previous choices (a and c) to the main figure (new Figure 5). This figure now also includes a group split, keeping with the new structure and the new focus on individual differences throughout. We also complete the figure by including the motor beta lateralization time courses for comparison. We left the full-group time courses, and the complementary two panels showing the effects of current/previous stimulus categories, in the Supplement. We felt that it would otherwise overload the main figure, and the data show that previous choice, but not stimuli, are encoded in both IPS signals.

Thank you again for insightful and constructive comments.

Point-by-point reply to reviewer #3

In this paper, Urai & Donner use human MEG to investigate oscillatory power during perceptual choices, with the goal of disentangling neural signals related to choice and motor history, and how any biases impact on the (computational) decision process. They find oscillatory signals relating to choice and motor action, and differential roles in the evidence accumulation process.

While these are generally interesting results, I had some reservations.

We are pleased to hear that you find the results interesting.

1. If a proximal goal is simply to disentangle whether previous motor action or previous decisions decision bias future choice, I am unsure why that would require neural-level data. One could simply use variable (within participants) response mappings such as to determine whether people repeat their choice or their motor action? Are there no previous studies that have done this?

Previous behavioral studies (which we cite) have indeed dissociated the effects of previous motor actions and previous decisions: (Akaishi et al., 2014)(their Figure 2A), ourselves (Braun et al., 2018) and (Feigin et al., 2021) have later drawn the same conclusions using different behavioral manipulations and analysis/modeling approaches. We now elaborate on this previous body of work in more detail.

That said, we would like to clarify that the proximal goal of the present study was not to dissociate the impact of decision and action per se on history bias. Rather, our goal was to dissect the large-scale neural implementation of choice history bias, by determining and comparing the functional impact of a range of visual, parietal, and frontal cortical regions, all of which have been implicated in the encoding of sensory evidence and/or choice and/or history bias by previous work. The key point here is that previous studies of choice history bias have typically focused on a single brain region. In addition, our approach afforded two further key insights beyond those from previous neural work on choice history bias: (i) linking these neural history signals to individual differences in choice history bias, which are pervasive (Urai et al. 2019) but understudied; and (ii) moving from simple identification of ‘neural correlates’ of history towards specifying their functional significance for behavior, by means of behavioral modelling. We have revised the Introduction and summary paragraphs of Discussion to clarify these points.

2. If both choice and motor action bias the evidence accumulation process, how can it be explained that only one of them biases the ultimate decision on the next trial?

This result may seem counterintuitive, but can be understood in the context of the dynamics of the DDM. Specifically, starting point adds to the offset of the integration, whereas drift bias is accumulated alongside the sensory evidence – its effect growing with time. As a result, simulations (reproduced below from Urai et al. 2019) show that the effect of starting point is only present on very fast RTs, whereas drift bias affects choice patterns across all trials. On the majority of trials, the fast (negative) effect of starting point) will thus quickly be masked by the (slower, integrative) effect of drift bias that then ultimately dominates choices.

Figure R2. Two biasing mechanisms within the DDM. The DDM postulates that noisy sensory evidence is accumulated over time, until the resulting decision variable y reaches one of two bounds (dashed black lines at $y = 0$ and $y = a$) for the two choice options. Repeating this process over many trials yields RT distributions for both choices (plotted above and below the bounds). Gray line: example trajectory of the decision variable from a single trial. Black lines: mean drift and resulting RT distributions under unbiased conditions. (a) Choice history-dependent shift in starting point. Green lines: mean drift and RT distributions under biased starting point. Gray-shaded area indicates those RTs for which starting point leads to choice bias. (b) Choice history-dependent shift in drift bias. Blue lines: mean drift and RT distributions under biased drift. Gray shaded area indicates those RTs for which drift bias leads to choice bias. (c) Both mechanisms differentially affect the shape of RT distributions. Conditional bias functions (White and Poldrack, 2014), showing the fraction of biased choices as a function of RT, demonstrate the differential effect of starting point and drift bias shift. Reproduced from Urai et al. 2019 *eLife*

3. In the results, it was often not clear to me what statistical test a statement was based on, and how each result accounted for the number of simultaneous tests involved (e.g. sensors, time and frequency windows, regions of interest). Given the multitude of reported results (and of perhaps equally plausible analyses), it might be useful to point out how their number is accounted for.

We understand what you refer to with “multitude of reported results”: we show a lot of data. But we account for multiple comparisons throughout figure panels, and many of the results shown are either control analyses (to strengthen our conclusions) or replications of previously established findings (to verify the quality of our data). The novel effects that are key for our present conclusions are the results presented in Figures 4b-d and f-g, Figure 5a, Figure 6, and Figure 7. Of those, only Figures 4 and 5 involved multiple comparisons across ROIs (Figure 4) or across time points (Figure 5). As specified in the figure legends, we used FDR-correction to control for multiple comparisons in each of them.

We would like to highlight that our selection of the frequencies, cortical areas, and time windows of interest featuring in these analyses was entirely hypothesis-driven, based on the results from our extensive previous MEG work on large-scale neural signatures of sensory evidence encoding, evidence accumulation, and motor preparation in random dot motion decision-making tasks similar to our present one (Siegel et al., 2006, 2011; Donner et al., 2009). So, in fact, our initial MEG figures (Figures 2 and 3), are complete replications of these established findings. The same could be said for the new figure characterizing behavioral sequences, replicating effects previously reported by ourselves (Urai et al., 2017) and others (Akaishi et al., 2014).

We present these figures to verify the quality of our data; and we use statistical assessment in line with common conventions (again correcting for multiple comparisons, in Figures 2 and 3, in fact across time and frequency using a cluster-based permutation procedure). We could have just derived all these replicated effects from the literature without replicating them here, select the relevant data features of interest and present the main findings in Figures 4-7 right away. But again, we feel that presenting this replication of established physiological signals, which is a widely applied standard in the systems neuroscience literature, strengthens the confidence in our findings. The replications are valuable in their

own right in light of the current replication crisis. There may be alternative analysis approaches, but ours is principled and hypothesis-driven, and follows directly from that previous work.

Your comment made us realize that we have failed to bring this general logic of our MEG approach (the separation between replication and novel effects) across sufficiently clearly in the previous version of the manuscript. We have now done this at the corresponding position in Results, before diving into the MEG findings (p. 4, 2nd par from bottom):

“To pinpoint neural signatures of choice history bias in our task, we focused on established MEG signatures of visual motion processing and action planning, within well-defined cortical regions and frequency bands (Siegel et al., 2006; Donner et al., 2009; Donner and Siegel, 2011; de Lange et al., 2013; Wilming et al., 2020; Murphy et al., 2021). We first replicated these signatures in our current measurements (Figures 2 and 3). This laid the ground for delineating, and functionally characterizing, choice history signals within the same areas and frequency bands (Figures 4-7).”

We hope that these clarifications and revisions are satisfactory.

4. The authors sometimes use FDR correction; unlike FWE this implies that 5% of reported results are expected to be false positives; this should be noted.

We have now noted this in a new Methods section ‘Statistics’.

5. The analysis has many degrees of freedom. Just to mention a few: the definition of frequency bands, of time windows for sensor selection and number of selected sensors, the definition of ROIs etc. How can the authors ensure that their statistical results are not due to particular settings of the variables?

We pre-selected those frequency bands and ROIs for which we have specific hypotheses, based on a large body of previous work that has mapped out the neural signatures of sensory evidence encoding and response preparation in tasks much like ours. Having confirmed these known effects, we then take these functionally-relevant neural signals and address our specific question of neural choice history bias, and its individual variability.

6. Some statements about the absence of effects are based on non-significance in a NHST; this interpretation is not supported by the test (particularly not if the p-value is slightly above .05: "The choice history effect in this area, correspondingly, did not interact with the effect of current stimulus category", p5 and other examples).

Agreed. We have now eliminated such statements.

7. What is the rationale of reporting confidence intervals? Is this a journal requirement? It might be useful to point out their value (see e.g. (Morey et al., 2016) for why they are or can be misleading).

We do not have strong feelings regarding the value of confidence intervals, we just used them in a rather conventional way. We now show 95 CIs as error bars throughout the paper, to harmonize the visualization of effect sizes from our general linear mixed models (where individual data points are not available) and individually-fit behavioral models. In the Methods section, we further clarify this to avoid confusion.

Thank you again for your insightful comments.

References

- Akaishi R, Umeda K, Nagase A, Sakai K (2014) Autonomous Mechanism of Internal Choice Estimate Underlies Decision Inertia. *Neuron* 81:195–206.
- Akrami A, Kopec CD, Diamond ME, Brody CD (2018) Posterior parietal cortex represents sensory history and mediates its effects on behaviour. *Nature* 554:368–372.
- Braun A, Urai AE, Donner TH (2018) Adaptive history biases result from confidence-weighted accumulation of past choices. *J Neurosci* 38:2418–2429.
- de Lange FP, Rahnev DA, Donner TH, Lau H (2013) Prestimulus Oscillatory Activity over Motor Cortex Reflects Perceptual Expectations. *J Neurosci* 33:1400–1410.
- Desender K, Boldt A, Verguts T, Donner TH (2019) Confidence predicts speed-accuracy tradeoff for subsequent decisions. *eLife* 8:e43499.
- Donner TH, Siegel M (2011) A framework for local cortical oscillation patterns. *Trends Cogn Sci* 15:191–199.
- Donner TH, Siegel M, Fries P, Engel AK (2009) Buildup of Choice-Predictive Activity in Human Motor Cortex during Perceptual Decision Making. *Curr Biol* 19:1581–1585.
- Feigin H, Baror S, Bar M, Zaidel A (2021) Perceptual decisions are biased toward relevant prior choices. *Sci Rep* 11:648.
- Fründ I, Wichmann FA, Macke JH (2014) Quantifying the effect of intertrial dependence on perceptual decisions. *J Vis* 14:9–9.
- Lak A, Hueske E, Hirokawa J, Masset P, Ott T, Urai AE, Donner TH, Carandini M, Tonegawa S, Uchida N, Kepecs A (2020) Reinforcement biases subsequent perceptual decisions when confidence is low, a widespread behavioral phenomenon. *eLife* 9:e49834.
- Machens CK, Romo R, Brody CD (2005) Flexible control of mutual inhibition: a neural model of two-interval discrimination. *Science* 307:1121–1124.
- Mendonça AG, Drugowitsch J, Vicente MI, DeWitt EEJ, Pouget A, Mainen ZF (2020) The impact of learning on perceptual decisions and its implication for speed-accuracy tradeoffs. *Nat Commun* 11:2757.
- Murphy PR, Wilming N, Hernandez-Bocanegra DC, Prat-Ortega G, Donner TH (2021) Adaptive circuit dynamics across human cortex during evidence accumulation in changing environments. *Nat Neurosci* 24:987–997.
- Pape A-A, Siegel M (2016) Motor cortex activity predicts response alternation during sensorimotor decisions. *Nat Commun* 7:13098.
- Purcell BA, Kiani R (2016) Neural Mechanisms of Post-error Adjustments of Decision Policy in Parietal Cortex. *Neuron* 89:658–671.
- Romo R, Salinas E (2003) Flutter Discrimination: neural codes, perception, memory and decision making. *Nat Rev Neurosci* 4:203–218.
- Siegel M, Donner TH, Oostenveld R, Fries P, Engel AK (2006) High-Frequency Activity in Human Visual Cortex Is Modulated by Visual Motion Strength. *Cereb Cortex* 17:732–741.
- Siegel M, Engel AK, Donner TH (2011) Cortical Network Dynamics of Perceptual Decision-Making in the Human Brain. *Front Hum Neurosci* 5.
- Urai AE, Braun A, Donner TH (2017) Pupil-linked arousal is driven by decision uncertainty and alters serial choice bias. *Nat Commun* 8:14637.
- Urai AE, de Gee JW, Tsetsos K, Donner TH (2019) Choice history biases subsequent evidence accumulation. *eLife* 8:e46331.
- White CN, Poldrack RA (2014) Decomposing bias in different types of simple decisions. *J Exp Psychol Learn Mem Cogn* 40:385–398.
- Wilming N, Murphy PR, Meyniel F, Donner TH (2020) Large-scale dynamics of perceptual decision information across human cortex. *Nat Commun* 11:5109.
- Zhang H, Alais D (2020) Individual difference in serial dependence results from opposite influences of perceptual choices and motor responses. *J Vis* 20:2–2.

REVIEWERS' COMMENTS

Reviewer #1 (Remarks to the Author):

The authors addressed all of my concerns.

Reviewer #2 (Remarks to the Author):

The authors did an excellent job at addressing all my suggestions. I commend the authors for their efforts and recommend the manuscript for publication.

Reviewer #3 (Remarks to the Author):

Thank you for this revision, which addresses my previous concerns.